# Dynamic pathways in energy landscapes guiding supramolecular Janus dendrimer self-assemblies between lamellar and cubic architectures

Jiabin Luan [1], Danni Wang [1], Niels P. Kok[1], Neshat Moslehi[2], Ilja K. Voets [2] & Daniela A. Wilson [1] ✉

Elaborate kinetic control enables supramolecular self-assemblies to deviate from equilibrium, depicting profound energy landscapes with remarkable structural and functional diversity. Despite this potential, achieving an energy landscape that encompasses lamellar and inverse cubic structures remains a significant challenge, contrasting with the sophisticated structural transformations naturally orchestrated by cellular systems. Here, we present a dynamic and minimalistic Janus dendrimer self-assembly system capable of reversibly transitioning between lamellar vesicles and inverse cubic structures. By exploiting temperature-triggered non-covalent interactions, including OEG interdigitation and hydrogen bonding, conformational flexibility, and molecular packing, we reveal a rich energy landscape featuring diverse assembly pathways spanning lamellar vesicles and inverse cubosomes. Our study not only enriches the structural versatility of Janus dendrimer assemblies but also provides a foundation for advancing supramolecular systems toward applications in biomedicine, catalysis, and beyond.

Self-assembly of molecules is a fundamental feature in nature and in technology[1]. Molecules are driven by intra- and intermolecular interactions to organize into higher order structures without further intervention[1,2]. Well-known examples are the self-assembly of lipids to form cell membranes and nanoparticles for mRNA vaccines against coronavirus disease 2019. The interest in self-assembly has catalyzed the development of supramolecular chemistry. In return, supramolecular chemistry advances the design of artificial cells, providing insights into the origins of cells[3,4], and accelerates the emergence of functional materials with unprecedented properties, including electronic and photic devices[5], active micro/nanomotors[6], and metabolizable supramolecular plastics[7], among others[8,9].

Relying on the molecular design of building blocks, molecules self-assemble into distinct hierarchical structures under thermodynamic equilibrium to meet various application demands. For amphiphilic molecules, such as natural lipids and synthetic polymers, the thermodynamically stable structure is primarily determined by the packing parameter, $p = v/a_0 l_c$, where $v$ represents the volume of the hydrophobic chains, $a_0$ is the surface area at the hydrophobic/hydrophilic interface, and $l_c$ is the length of the hydrophobic segment[10,11]. The packing parameter elegantly predicts the formation of spherical and cylindrical micelles ($p < 1/3$ and $1/3 < p < 1/2$, respectively)[12], vesicles ($1/2 < p < 1$)[12], planar lamellae ($p = 1$), and inverse structures ($p > 1$), such as cubosomes and hexosomes. More

[1]Institute for Molecules and Materials, Radboud University, Nijmegen, The Netherlands. [2]Laboratory of Self-Organizing Soft Matter, Department of Chemical Engineering and Chemistry and Institute of Complex Molecular Systems, Eindhoven University of Technology, Eindhoven, The Netherlands. ✉e-mail: d.wilson@science.ru.nl

compelling and consistent with natural processes, self-assembly systems do not conform to a single static equilibrium state. Instead, molecules can self-assemble into a variety of structures as kinetically trapped states. These out-of-equilibrium energy minima, along with the thermodynamically equilibrium state, draw an energy landscape via the same building blocks of molecules[13–15]. Embedded within this landscape are distinct structures and functions, which drive the adaptivity and diversity observed in cells. This is exemplified by the remarkable morphological variability of exosomes for cellular communication[16]. By fine-tuning the balance between antagonistic non-covalent interactions, we can now steer supramolecular assemblies across complex energy landscapes, as demonstrated in one-dimensional fibers[17,18], two-dimensional sheets[19], and more recently, three-dimensional vesicles[20]. However, exploring the full complexity of landscapes in three-dimensional realms remains limited as compared to the sophisticated structures that cells can naturally achieve. While cell membranes are typically organized as flat bilayers, they can also adopt highly curved, three-dimensional periodic structures known as cubic membranes[21]. These nonlamellar structures, including inverse cubic ($Q_{II}$) or hexagonal ($H_{II}$) phases, are extensively observed in endoplasmic reticulum and mitochondrial inner membranes, often in response to cellular stress[22], disease[23], starvation[24], or viral infection[25].

To investigate the formation and function of these intriguing non-lamellar cubic structures, researchers have accomplished various inverse cubic phases using complex lipid mixtures by incorporating monoolein and/or PEG-based stabilizer[26–29], modifying the structure and length of synthetic polymers[30,31], and altering preparation conditions such as solvent type and concentration[32,33]. Despite these advances, the formation of inverse cubic phases remains challenging as demonstrated by their sensitive phase behavior to the subtle presence of impurities[34]. The intricate molecular compositions required, coupled with the non-trivial "ideal" preparation conditions, present significant obstacles achieving a diverse energy landscape with both lamellar and cubic structures. Here, we address this challenge by employing a single-molecular self-assembly system to plot a rich energy landscape that demonstrates reversible transitions between lamellar and inverse cubic structures. We design our self-assembled system using Janus dendrimer molecules, which offer great chemical functionality, stability, and flexibility[35,36]. In our previous study, self-assemblies using Janus dendrimers, which composed of (3,4)-patterned hydrophobic aliphatic chains and (3,5)-patterned hydrophilic oligo(ethylene glycol) (OEG) on phenolic acid units, exhibited an energy landscape featuring various types of vesicles[20]. Temperature served as a switch, modulating the balance between OEG interdigitation and the encoded conformational freedom, thereby guiding the pathway selection of distinct vesicles. The vesicle-dominated landscape suggested a packing parameter ($p$) between 0.5 and 1[10,11]. We hypothesize that modifying $p$ to exceed 1 could potentially expand the energy landscape into the realm of inverse cubic structures.

In this work, we invert the positions of hydrophobic and hydrophilic units: aliphatic chains were placed in (3,5) position, while OEG occupied the (3,4) position. This molecular structural design increases the hydrophobic chain volume ($v$) due to (3,5)-branching and decreases the hydrophilic headgroup area ($a_0$) of OEG as a result of the (3,4)-branching pattern. Compared to the previous molecule, the current design displays a larger packing parameter ($p > 1$) (Fig. 1a). By leveraging the interplay among OEG interdigitation, conformational freedom, and molecular packing, we envision that an energy landscape featuring unique pathway selections between lamellar vesicles and inverse cubic structures can be drawn (Fig. 1b).

## Results and discussion
### Kinetic and thermodynamic study of assemblies in the absence of ethanol
The self-assembly of Janus dendrimer, $(3,5)12G1\text{-}PE\text{-}(3,4)\text{-}3EO\text{-}G1\text{-}(OCH_3)_4$ (Supplementary Figs. 1 and 3), was initiated using an injection method. As shown in our previous work[20], the presence of ethanol, a good solvent for dendrimer, can shift the system to an alternative energy state. We first explored the energy landscape of self-assemblies in the absence of ethanol by completely removing ethanol through extensive dialysis (residual ethanol concentration below 4.8 ppm).

The rapid injection process does not allow the assemblies to reach their thermodynamic equilibrium state, resulting in kinetically trapped states. To return the system to equilibrium, heating was applied to provide an energy trigger. The evolution of self-assemblies, including hydrodynamic diameter ($D_h$), polydispersity index (PDI), and derived count rates, was monitored using light scattering (LS). Notable changes in the first heating trend were observed at 37 °C and 50 °C in both PDI and count rates, suggesting potential transitions between different energy states (Supplementary Fig. 4 and related discussion). To further investigate the annealing effects on the self-assemblies at 37 °C and 50 °C (heating to 37 °C or 50 °C for certain periods of time, then cooling down to room temperature (R.T.)), a range of analytical techniques was employed (Fig. 2a). Upon equilibrating at 37 °C for 24 h, the assemblies back to R.T. exhibited a gradual size increase of approximately 30% accompanied by a decrease in count rates, as observed by LS (Fig. 2b). Concurrently, the PDI showed a significant reduction of over 50%, indicating a shift toward more well-defined assemblies. These parameters reached a plateau after annealing for 24 h and remained constant through 72 h. Similar trends in relative changes in the size and particle concentration were corroborated by nanoparticle tracking analysis (NTA) during the annealing treatment at 37 °C (Fig. 2c). These results suggest that particle aggregation and/or fusion occur, leading to a decrease in particle concentration. To further elucidate the nature of these structural variations, and in particular, whether the annealing process brings the system to different energy states, the morphologies of self-assemblies during kinetic annealing at 37 °C were closely examined using cryogenic-transmission electron microscopy (cryo-TEM).

Similar to its counterpart, $(3,4)12G1\text{-}PE\text{-}(3,5)\text{-}3EO\text{-}G1\text{-}(OCH_3)_4$[20], the current molecule self-assembles into nonconcentric multivesicular vesicles (MVVs, also known as vesosomes) immediately after the self-assembly (Fig. 2d and Supplementary Fig. 5a). Upon annealing at 37 °C for 10 min, the MVVs undergo a surprising transition into an inverse sponge structure encapsulated within bilayers. These structures display characteristic pores, representing water channels (Fig. 2e and Supplementary Fig. 5b). Known as interlamellar attachments, these pores are typical intermediate structures formed during the transition from lamellar membranes to cubosomes via membrane fusion[27,37]. Complex vesicular-sponge structures, several micrometers in size, were observed (Supplementary Fig. 5c), which clearly shows evidence of fusion among multiple MVVs. As the annealing time proceeds to 9 and 24 h, vesicular-sponge structures dominate over MVVs (Fig. 2f, g and Supplementary Fig. 5d, e). Fast Fourier transform (FFT) analysis of the sponge phase reveals a disordered feature, confirming the absence of periodicity (Fig. 2i). In the meantime, cubosomes, characterized by structural periodicity in FFT analysis, gradually emerged (Fig. 2j). This indicates the sponge structure serves as an intermediate state during the transition from MVVs to cubosomes. This statement is further substantiated as cubosomes became the most prominent structures after 72 h of annealing (Fig. 2h and Supplementary Fig. 5f). To investigate the transition kinetics of different types of assemblies, we quantified the populations of various structures by randomly imaging a substantial number of particles across different areas and batches (>500 particles per condition). This approach aimed to closely reflect the true state of the samples[20]. Throughout this study, we employed this protocol to monitor sample morphologies and gain insights into the kinetics of morphological transitions (Fig. 2k). In the initial self-assembly, MVVs are the dominant type of the sample with nearly 90% of the population, followed by unilamellar vesicles (ULVs) contributing approximately 10% and oligolamellar vesicles (OLVs) forming a

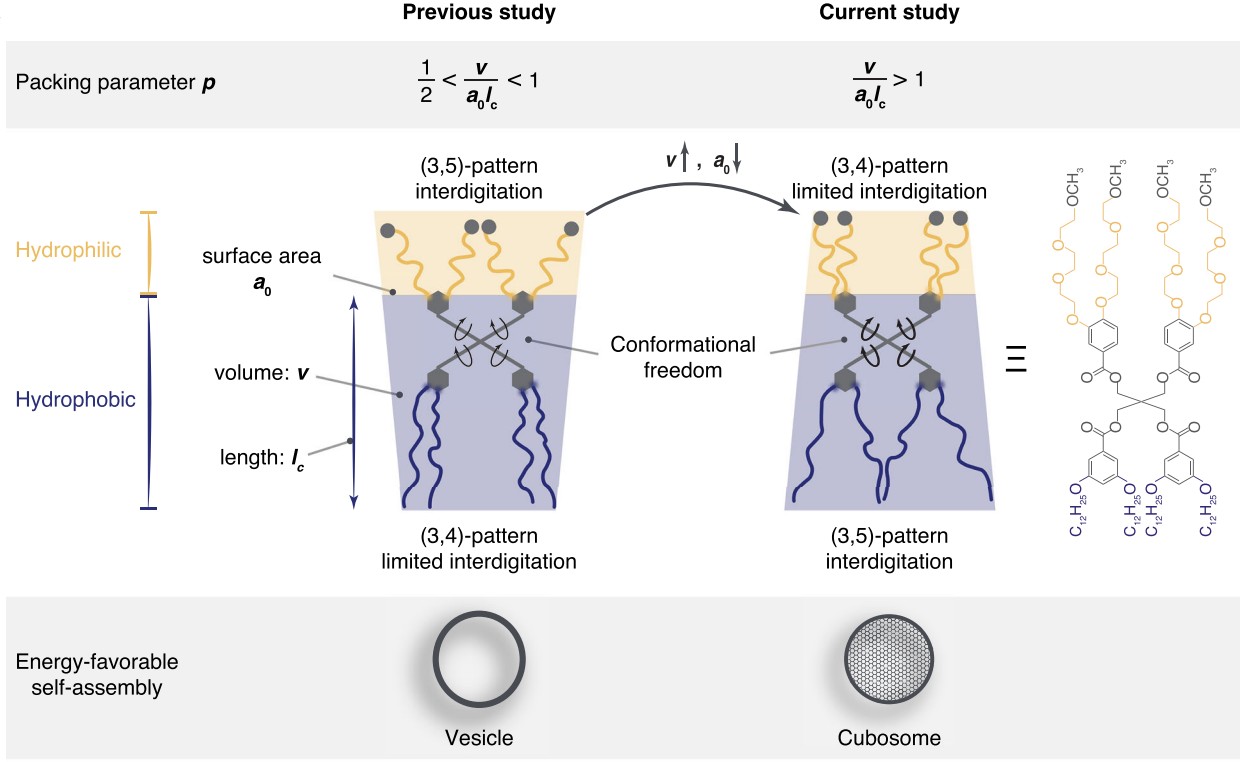

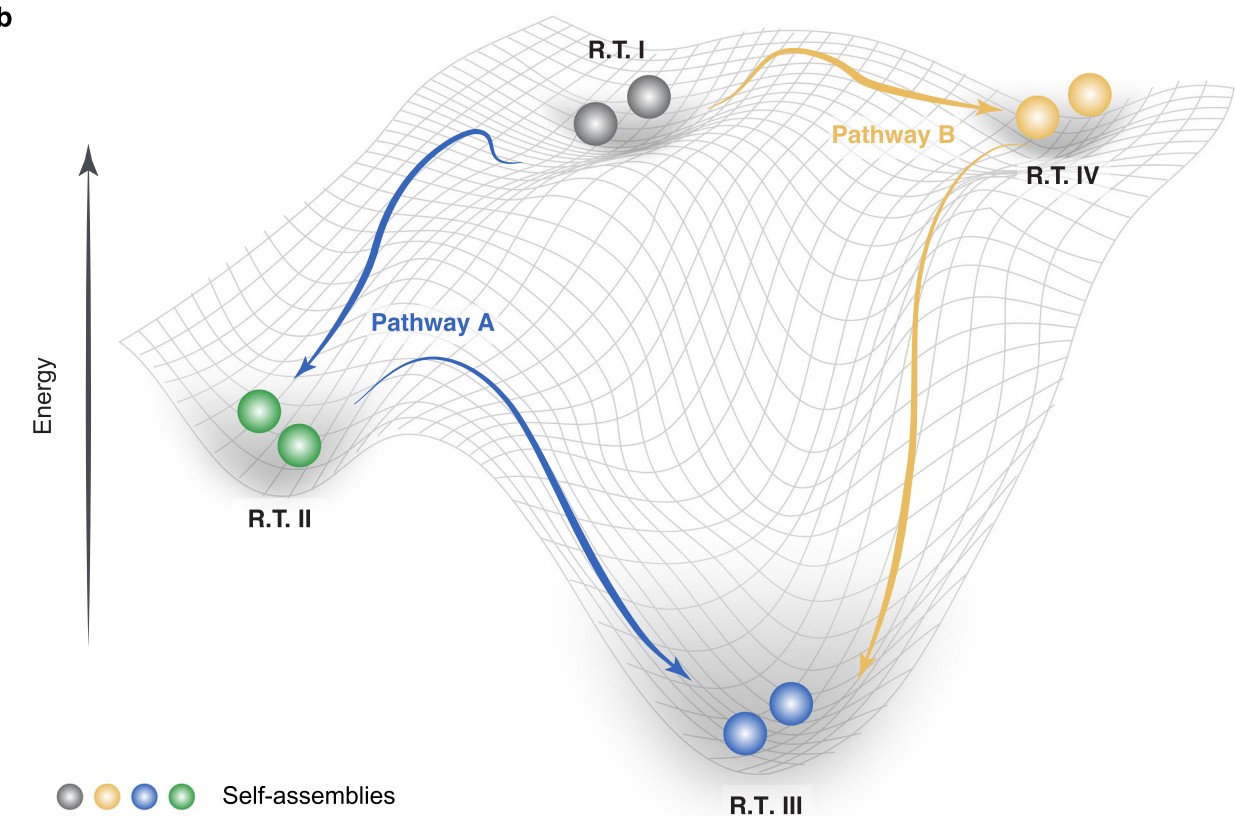

**Fig. 1 | Energy landscapes featuring unique pathway selections of Janus dendrimer self-assemblies. a** Design of molecular building blocks with increased packing parameter for energy-favorable inverse cubic structures. **b** Free energy landscapes of Janus dendrimer assemblies with distinct pathway selections, leading to multiple energy states (I–IV) at room temperature (R.T.).

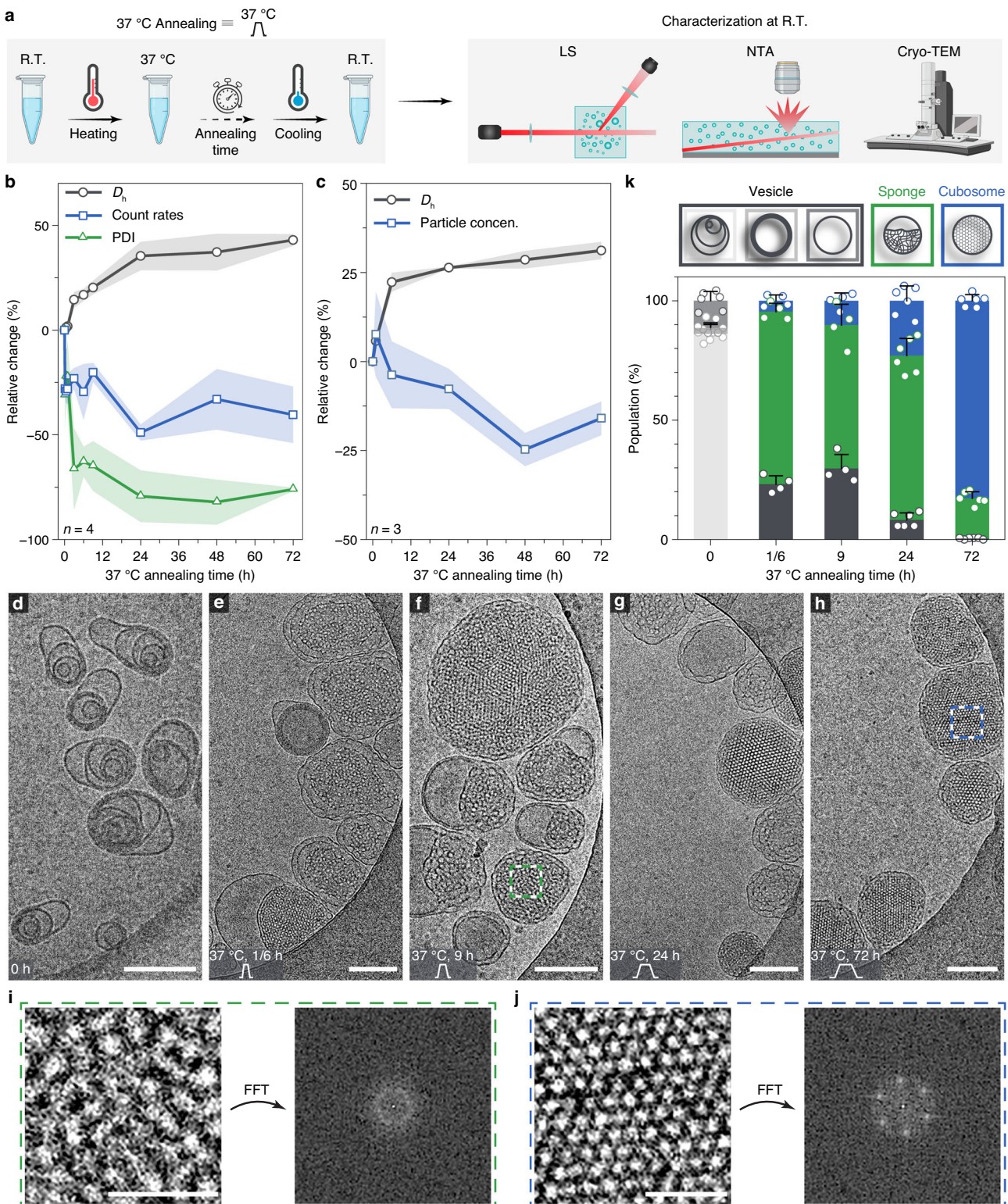

**Fig. 2 | Kinetic investigation of self-assembled Janus dendrimers via 37 °C annealing in the absence of ethanol. a** Annealing process at 37 °C and characterization of self-assemblies. Elements created in BioRender. Wilson, P. (2025) https://BioRender.com/djwnhzu. **b**, **c** Relative changes of $D_h$ (%), PDI (%), and derived count rates (%) to the starting points as measured by LS (**b**), and relative changes of $D_h$ (%) and particle concentration (particle concen.) (%) to the starting points as measured by NTA (**c**) as a function of annealing time at 37 °C. Measurements were performed on multiple batches of samples, with standard deviation (s.d.) presented in shaded areas ($n = 4$ for LS; $n = 3$ for NTA). **d–h** Cryo-TEM images

of self-assemblies following ethanol removal (**d**) and assemblies annealed at 37 °C for 10 min (**e**), 9 h (**f**), 24 h (**g**), and 72 h (**h**). Scale bars are 200 nm. **i**, **j** Fast Fourier transformations (FFT) analysis of selected zoom-in regions in (**f**) and (**h**), showing a disordered sponge structure without periodicity (**i**) and an ordered cubosome structure with periodicity (**j**). Scale bars are 50 nm. **k** Quantitative analysis of assembly populations after ethanol removal and post-annealing at 37 °C. For each condition, images from multiple areas and batches were analyzed to minimize the error ($n > 500$ particles per condition). Bar plots represent mean ± s.d.

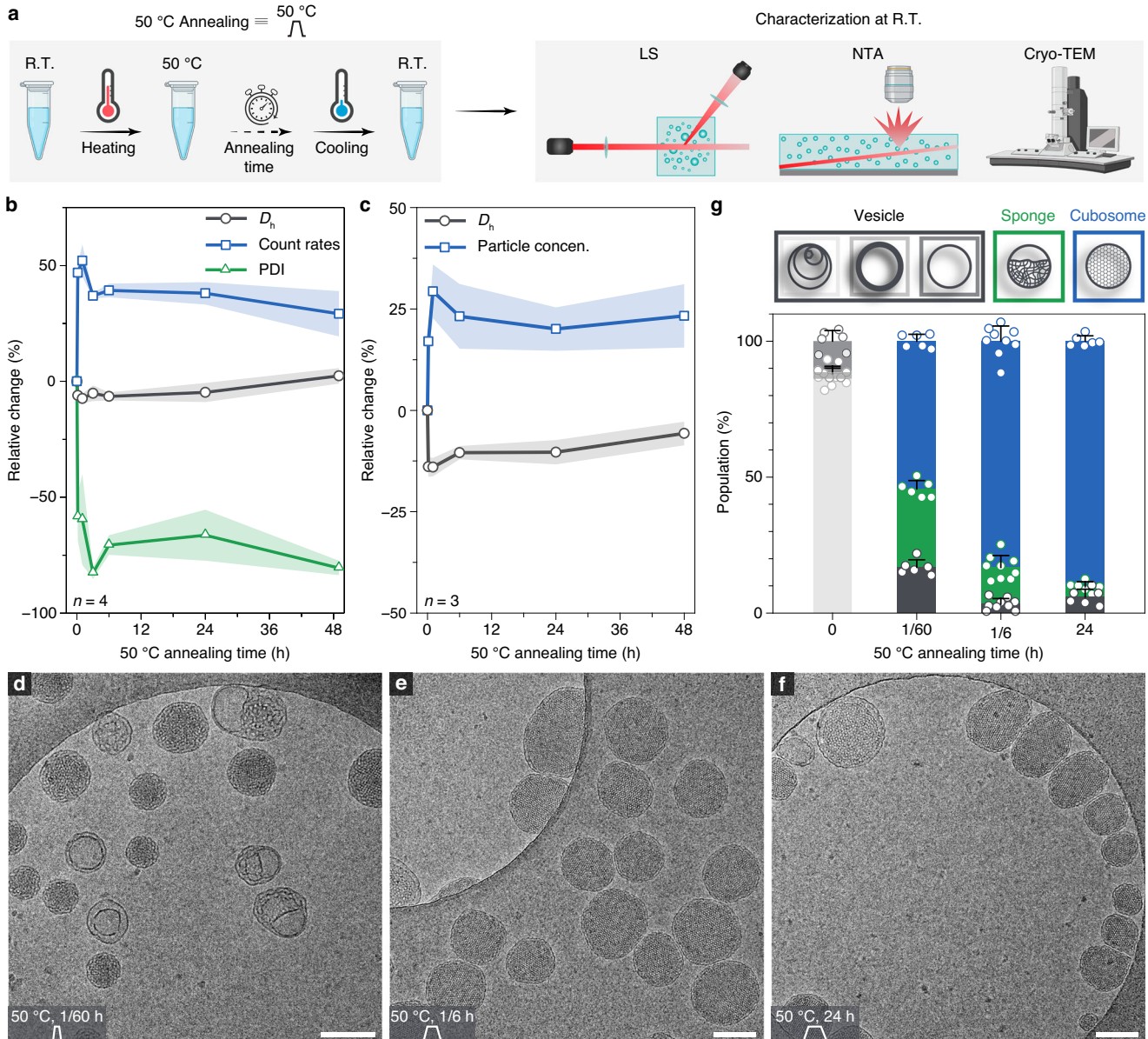

**Fig. 3 | Kinetic investigation of self-assembled Janus dendrimers via 50 °C annealing in the absence of ethanol. a** Annealing process at 50 °C and characterization of self-assemblies. Elements created in BioRender. Wilson, P. (2025) https://BioRender.com/djwnhzu. **b, c** Relative changes of $D_h$ (%), PDI (%), and derived count rates (%) to the starting points as measured by LS (**b**), and relative changes of $D_h$ (%) and particle concentration (%) to the starting points as measured by NTA (**c**) as a function of annealing time at 50 °C. Measurements were performed on multiple batches of samples, with s.d. presented in shaded areas ($n = 4$ for LS; $n = 3$ for NTA). **d–f** Cryo-TEM images of self-assemblies annealed at 50 °C for 1 min (**d**), 10 min (**e**), and 24 h (**f**). Scale bars are 200 nm. **g** Quantitative analysis of assembly populations after ethanol removal and post-annealing at 50 °C. For each condition, images from multiple areas and batches were analyzed to minimize the error ($n > 600$ particles per condition). Bar plots represent mean ± s.d.

marginal percentage. Annealing at 37 °C triggered a dramatic morphological shift, marked by a significant decrease in vesicles. For simplicity, we grouped all vesicle types (MVVs, OLVs, and ULVs, Supplementary Fig. 6) into a single category as Vesicle. After 10 min to 24 h of annealing, the intermediate vesicular-sponges emerged as the predominant structure, representing 60–70% of the population. Simultaneously, the percentage of cubosomes increased gradually, from 5% to 20%, reflecting their slower transition kinetics. Upon annealing for 72 h, the transition to cubosomes was nearly complete, with cubosomes exceeding 80% of the population. Prolonged annealing (156 h) showed no further significant increase in cubosome prevalence (Supplementary Fig. 7).

We next explored the effects of the other identified temperature, 50 °C, on the energy states of the self-assemblies (Fig. 3a). It was

anticipated that the system would achieve its thermodynamically stable state more rapidly compared to annealing at 37 °C. While the PDI measured by LS displayed a similarly significant decrease after annealing at 50 °C, comparable to the result observed at 37 °C, the size and count rate trends differed markedly (Fig. 3b). Specifically, the size of the assemblies decreased moderately by approximately 10% relative to the original sample, and the count rates and particle concentrations increased by the range of 25–50%, as demonstrated by both LS and NTA (Fig. 3b, c).

Without morphological information, further comparison with the results from annealing at 37 °C was not possible. To address this, cryo-TEM was employed to monitor morphological transitions during the annealing process at 50 °C. Similar to the transitions observed at 37 °C, vesicles gradually transformed into intermediate sponges and then

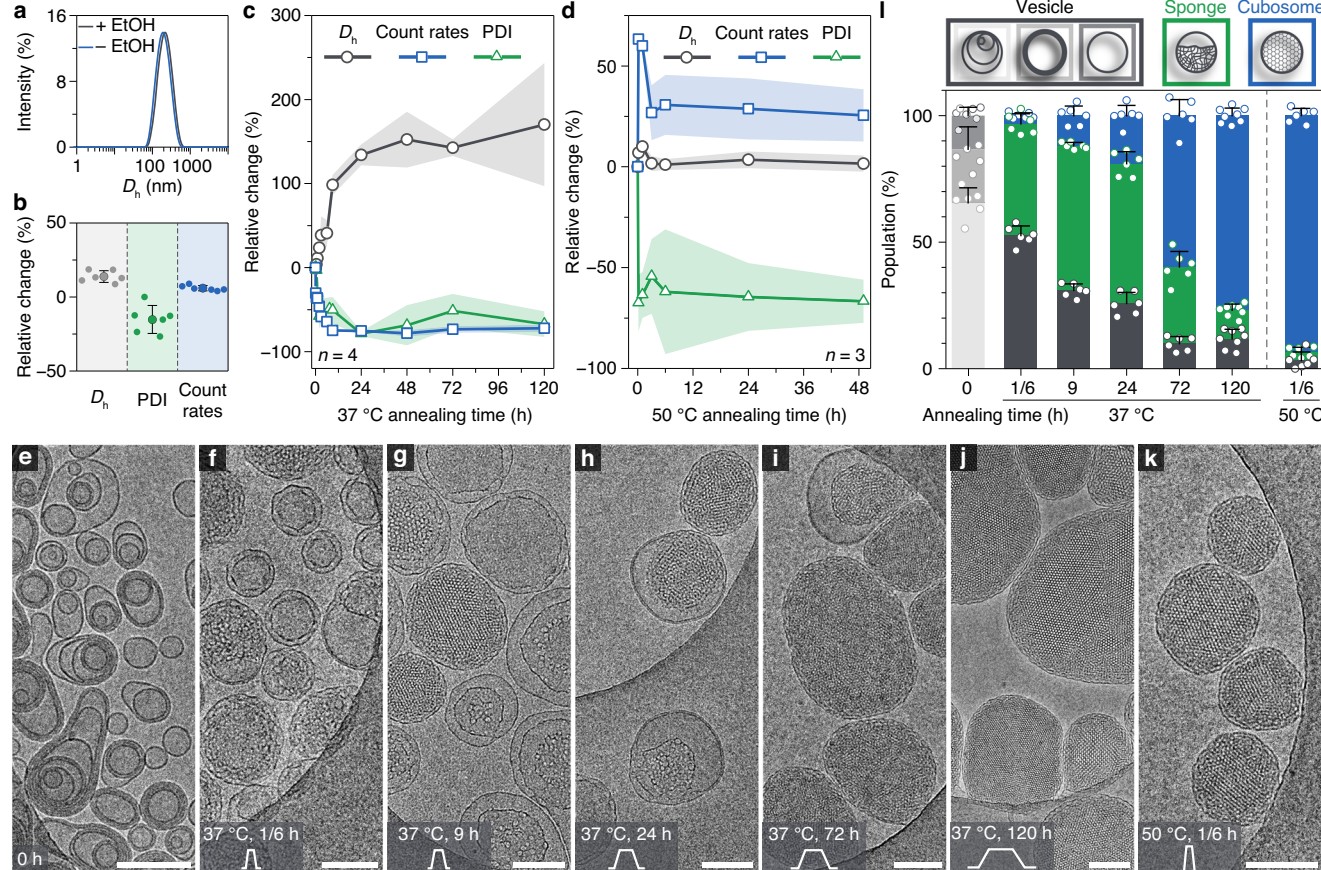

**Fig. 4 | Kinetic investigation of self-assembled Janus dendrimers via 37 °C and 50 °C annealing in the presence of ethanol. a** $D_h$ profiles determined by LS in the presence (+EtOH) and absence (−EtOH) of ethanol. **b** Relative changes of $D_h$ (%), PDI (%), and derived count rates (%) compared to the sample without ethanol, as measured by LS. Data are presented as mean ± s.d. **c**, **d** Relative changes of $D_h$ (%), PDI (%), and derived count rates (%) to the starting points as a function of annealing time at 37 °C (**c**) and 50 °C (**d**), respectively. Measurements were performed on multiple batches of samples, with s.d. presented in shaded areas ($n = 4$ for 37 °C;

$n = 3$ for 50 °C). **e−k** Cryo-TEM images of self-assemblies prepared by direct injection with ethanol (**e**) and assemblies annealed at 37 °C for 10 min (**f**), 9 h (**g**), 24 h (**h**), 72 h (**i**), and 120 h (**j**), and assemblies annealed at 50 °C for 10 min (**k**). Scale bars are 200 nm. **l** Quantitative analysis of assembly populations from direct injection with ethanol and after annealing at 37 °C and 50 °C. For each condition, images from multiple areas and batches were analyzed to minimize the error ($n > 600$ particles per condition). Bar plots represent mean ± s.d.

into cubosomes (Fig. 3d–f). As expected, the transition kinetics were significantly faster at 50 °C than at 37 °C. Most vesicles were replaced by cubosomes and sponges within 1 min of annealing. The transition was completed after 10 min, with over 80% of cubosomes formed (Fig. 3g). The transformation from MVVs to cubosomes was further validated and analyzed by small angle X-ray scattering (SAXS) (Supplementary Fig. 8). This rapid transition to the state of cubosomes in the timescale of minutes was attributed to the high energy input. Interestingly, cubosomes formed by annealing at 37 °C ($D_h$, 250 nm) were nearly 50% larger than those formed at 50 °C in size ($D_h$, 170 nm), as measured by LS. This size difference seems to indicate a different pathway of the transition from MVVs to cubosomes. The formation of cubosomes at R.T. after annealing at 50 °C prompted us to explore the effect of a second annealing cycle. It turned out that no further changes in cubosome morphology were observed after a second annealing from either 37 °C or 50 °C, indicating that cubosomes are thermodynamically stable (Supplementary Fig. 9).

Given the sensitivity of self-assemblies to thermal history, we further examined the effects of self-assembly concentration and R.T. aging prior to annealing on morphological transitions. To assess the potential concentration effects, the self-assembly concentration was systematically varied from 0.5 to 1.0 and 0.1 mg/mL, respectively. The concentration changes had minimal effect on both the morphology of self-assemblies and the MVV-to-cubosome transition upon annealing

at 50 °C (Supplementary Fig. 10). Similarly, aging self-assemblies at R.T. for 3 days did not pose noticeable impact on the MVV structures as well as their transformation to cubosomes following 50 °C annealing treatment (Supplementary Fig. 11).

## Kinetic and thermodynamic study of assemblies in the presence of ethanol

We revisited the role of ethanol (approximately 5% by volume) in navigating the energy landscapes of the self-assemblies. The presence of ethanol resulted in a slight diameter increase from 175 nm to 200 nm, as measured by LS (Fig. 4a). This increase was attributed to the swelling effect of ethanol on the area per molecule, which was also observed in lipid molecules with similar alkyl chains[20,38,39]. Moreover, the original assemblies with ethanol displayed a slightly lower PDI, while the count rates remained constant (Fig. 4b). Overall, the self-assemblies exhibited similar monomodal peaks, with only moderate variations observed in batch measurements by LS. These moderate differences were further substantiated by cryo-TEM. Compared to the sample without ethanol (Fig. 2d and k), the percentage of MVVs, which constituted the major population, decreased from approximately 85% to 65% in the presence of ethanol (Fig. 4e and l). The reduction in MVVs was offset by an increased proportion of ULVs and OLVs (Fig. 4l).

The effect of annealing on the energy states was then carefully examined by annealing the samples at 37 °C and 50 °C, respectively.

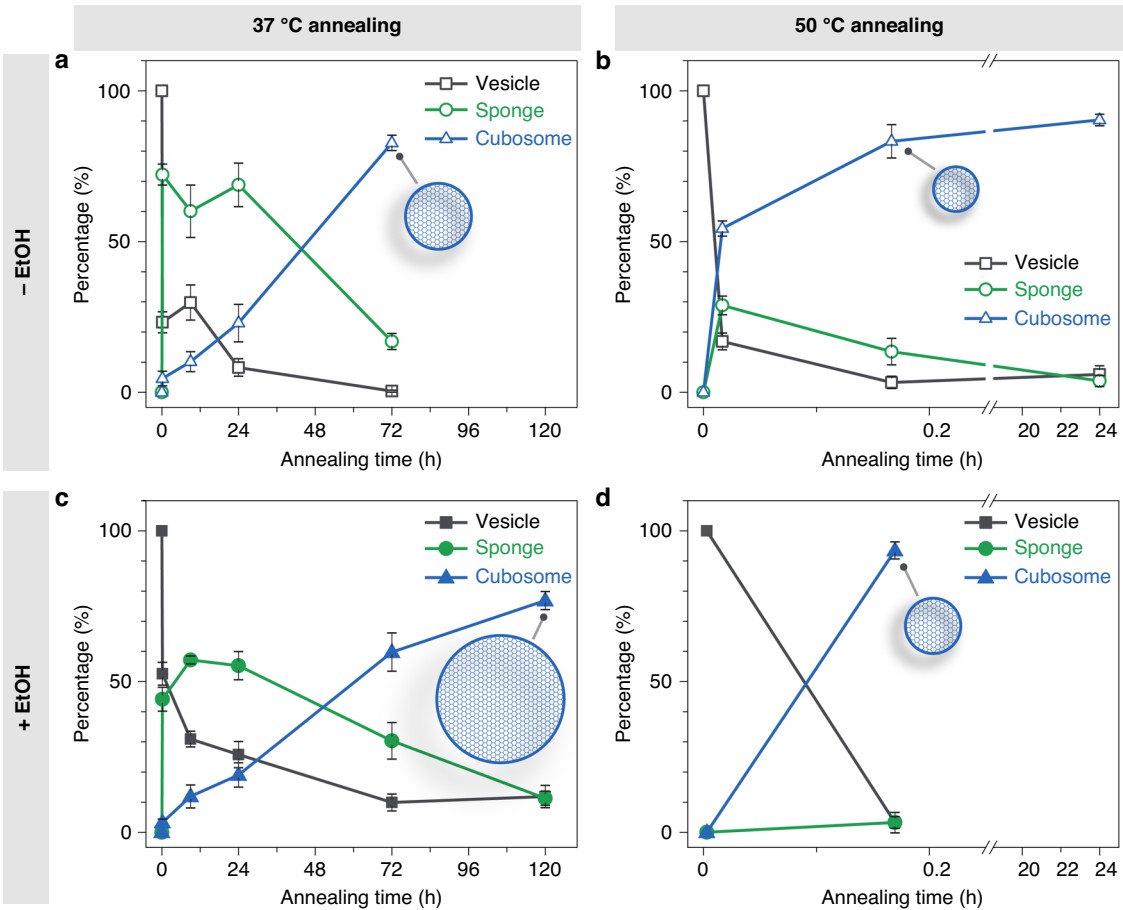

**Fig. 5 | Modulation of transition kinetics and dimensions of self-assembled Janus dendrimers by annealing temperature and solvent.** Modulated by temperature: **a**, **c** 37 °C; **b**, **d** 50 °C. Modulated by solvent: **a**, **b** without ethanol, −EtOH; **c**, **d** with ethanol, +EtOH. Data are presented as mean ± s.d. Cubosome particles of varying dimensions are shown upon completion of the transition for each condition.

Similar to the LS results for samples annealed at 37 °C in the absence of ethanol (Fig. 2b), the relative changes showed an increase in size and decreases in count rates and PDI (Fig. 4c). However, a notable difference was observed in the magnitude of size increase: at 37 °C, the size increased by 170% with ethanol, compared to only 43% without ethanol after 72 h of annealing. This variation in size change was not observed for samples annealed at 50 °C, where the presence or absence of ethanol produced nearly identical trends. Specifically, the size remained almost constant, the PDI decreased significantly, and the count rates increased moderately via 50 °C annealing (Fig. 4d). A detailed discussion comparing the kinetic changes in LS for samples with and without ethanol will be provided after presenting the corresponding morphological information, which we address next.

The kinetically trapped vesicles gradually transitioned to cubosomes via intermediate sponges during annealing at 37 °C, reaching 77% cubosomes formation over a prolonged duration of 120 h (Fig. 4e–j and l and Supplementary Fig. 12). Notably, in the presence of ethanol, the resulting cubosomes were significantly larger in size compared to those formed without ethanol, which is consistent with the LS results. When the annealing temperature was raised to 50 °C, the transition to cubosomes proceeded much more rapidly, with no noticeable size change compared to the original vesicles (Fig. 4k and Supplementary Fig. 13). After just 10 min of annealing at 50 °C, cubosomes accounted for 90% of the sample population (Fig. 4l).

The annealing temperature and solvent act as crucial modulators, controlling the transition kinetics and dimensions of the particles among different energy states of the self-assemblies (Fig. 5). The annealing temperature primarily determines the kinetics of the vesicle-

to-cubosome transition. High energy input from annealing at 50 °C completes the transition within 10 min (Fig. 5b and d), whereas annealing at 37 °C requires at least 72 h to finish the process, with the transition kinetics of cubosomes exhibiting a nearly linear profile (Fig. 5a and c). Meanwhile, the size of the thermodynamically stable cubosomes depends on both the annealing temperature and the solvent. Cubosomes formed upon annealing at 50 °C for 10 min exhibited sizes similar to the original vesicles (−6% without ethanol, Fig. 3b, and +7% with ethanol, Fig. 4d). In contrast, annealing at 37 °C for 72 h in the absence of ethanol resulted in cubosomes 43% larger in diameter compared to the original vesicles (72 h, Fig. 2b). This increase was even more pronounced in the presence of ethanol, where the diameter grew by 170% (120 h, Fig. 4c). The observed size differences are reflected in cubosome particles of various dimensions shown in Fig. 5. These results indicate that fusion occurred as the transformation proceeded during 37 °C annealing, with ethanol significantly enhancing the fusion process due to the enhanced membrane fluidity as shown in lipid systems[39]. Our findings highlight the pivotal role of thermal history and solvent in modulating the kinetics of the transitions and dimensions of the cubosomes, which is essential for optimizing their biomedical applications[40].

## Energy landscapes modulated by temperature

To further elucidate the effect of temperature and navigate energy states via different pathways, we monitored the morphologies of the assemblies using cryo-TEM in real-time during the annealing procedure. To eliminate solvent interference, all following experiments were performed after ethanol removal. The samples remained stable and

uniform throughout the investigated temperature range (Supplementary Fig. 4). Following our previously reported procedure[20], samples were vitrified at temperatures above R.T., enabling high-quality cryo-TEM imaging of sample states at elevated temperatures.

Starting from the lamellar MVVs at R.T. (R.T. I, Fig. 6a), the samples transitioned predominantly into sponges after heating at 37 °C for 10 min (Fig. 6b). By comparing these sponges to those formed through 37 °C annealing (Fig. 2e), it was evident that these structures were preserved after cooling back to R.T. from 37 °C (Fig. 2e and R.T. II, Fig. 6c). If we increased the temperature of the self-assembled sample to 50 °C for 10 min, the inverse sponges unexpectedly transitioned back to a lamellar phase, forming a distinct morphology of multilamellar vesicles (MLVs, also known as onion vesicles, Fig. 6d **ii** and Supplementary Fig. 6). An intermediate state featuring MLVs with ill-defined curvy bilayers was also observed (Fig. 6d **i**). The transition back to vesicular structures at 50 °C suggests that the packing parameter of the molecules favored a value for MLV formation (lamellar phase, $0.5 < p < 1$) above 50 °C. Upon cooling back to R.T. from 50 °C, the MLVs uniformly transformed to inverse cubosomes, comprising more than 80% of the population (Fig. 3g), and established a third state at R.T. (R.T. III, Figs. 6e and 3e).

All the above findings demonstrate that at R.T., the energy-favored packing parameter of the molecules exceeds unity, driving the transition from original lamellar MVVs (R.T. I) to cubosomes (R.T. III). Interestingly, annealing at 37 °C and 50 °C led to two distinct transformation pathways from MVVs to cubosomes. In the first pathway (pathway **A**), annealing MVVs (lamellar phase) at 37 °C resulted in a gradual transformation into sponges and subsequently cubosomes (inverse phase, at both 37 °C and R.T.). In contrast, in the second route (pathway **B**), annealing at 50 °C produced MLVs (lamellar phase) at 50 °C from the inverse structures, which then converted into cubosomes upon cooling to R.T. The key distinction between these pathways lies in the formation of MLVs at 50 °C, a vesicular lamellar phase not observed at 37 °C. To further define the temperature threshold for pathway selection, we examined two annealing temperatures, 44 °C and 60 °C, below and above 50 °C, respectively, to monitor the real-time transitions throughout the annealing process. Annealing at 44 °C followed pathway **A** with inverse sponges and cubosomes observed (Supplementary Fig. 14a, b), whereas annealing at 60 °C followed pathway **B**, featuring MLV formation at 60 °C and cubosomes upon returning to R.T. (Supplementary Fig. 14c, d). Together, these results identify 50 °C as a critical transition temperature that governs the pathway selection from MVVs to cubosomes. To further quantify the energy barrier of the MVV-to-cubosome in pathway **A**, we performed cryo-TEM analysis across the temperature range of 37–44 °C. By means of an Arrhenius plot, the energy barrier for the transition was calculated to be 109 kJ mol$^{-1}$ (Supplementary Fig. 15).

We next sought to map the energy landscape of self-assemblies at R.T. with distinct morphologies by kinetically trapping the structures en route to the thermodynamically stable cubosomes. Sponges, which act as precursors to cubosomes, were readily trapped by interrupting the annealing process at 37 °C (pathway **A**), owing to the slow kinetics of the transition (Fig. 2k). As a kinetic intermediate, the Sponge state (R.T. II, Fig. 6g) occupies an energy minimum between the initial MVVs (R.T. I, Fig. 6g) and final Cubosome state (R.T. III, Fig. 6g). In contrast, MLVs only emerged above 50 °C suggesting a higher energy cost (pathway **B**), as reflected by the difficulty in kinetically trapping these structures at R.T. (Supplementary Fig. 16). The transition to cubosomes occurred consistently upon cooling from 50 °C, whether at a slow cooling rate of 1 °C/min (Supplementary Fig. 16a), a moderate cooling rate of 6 °C/min (Fig. 6e and Supplementary Fig. 16b), or via rapid air quenching to R.T. (Fig. 3e). We then employed a rapid quenching procedure from 50 °C to 0 °C using an ice-water bath. Upon return to R.T., the sample reverted to the same cubosome state (Supplementary Fig. 16c). This finding further confirms that cubosomes reside in a deep energy well and the molecules

are sufficiently dynamic to assemble into the thermodynamically stable cubosome state, even under rapid quenching at 0 °C. Subsequently, an ultrafast quenching procedure using liquid nitrogen was implemented to freeze the sample from 50 °C, followed by equilibration at R.T. This approach successfully trapped lamellar MLVs at R.T., resulting in a distinct fourth energy state (R.T. IV, Fig. 6f, g and Supplementary Fig. 16d). Notably, ultrafast quenching method enabled a transition from thermodynamic cubosomes back to kinetically trapped vesicular MLV structures, demonstrating a reversible transition between lamellar vesicles and inverse cubosomes. All four states remained stable over several days at R.T., indicating that each resides in a local energy minimum (Supplementary Fig. 16e and Supplementary Fig. 17).

Temperature serves as a pivotal switch for the packing parameters of molecules, dictating the self-assembled structures within the energy landscape (Fig. 6g). Two distinct pathways were identified, each leading to the formation of self-assembled structures, ranging from lamellar vesicles to inverse cubosomes, all originating from the same initial state. Our findings emphasize the importance of thermal history in formulation and highlight the potential for leveraging pathway selection to enable diverse applications or expand possibilities within the same application framework.

## Energy landscapes navigated by molecular design

The intriguing pathway complexity observed in the self-assemblies in Fig. 6g prompted further discussion regarding the molecular packing parameters that govern the final state of the structures. We expect that a deeper understanding of the behavior of these dynamic molecules will aid in achieving distinct self-assembly pathways by encoding molecular interactions in Janus dendrimer molecular design. Although the packing parameter favors the formation of energetically stable cubosomes (inverse phase, $p > 1$), kinetically trapped MVVs (lamellar phase, $0.5 < p < 1$) were formed during the fast injection self-assembly process (Fig. 2d and 4e). The formation of MVVs suggests that molecules are not able to arrange into the most stable packing state—inverse cubosomes—during this prompt injection process. A notable feature of the nonconcentric MVVs is the partial "intra-attraction" between bilayers, observed in the cryo-TEM images, regardless of the presence or absence of ethanol. This "intra-attraction" is a result of partial interdigitation of the OEG chains (Supplementary Fig. 18a–c). We propose two factors from a molecular perspective that contribute to this intriguing phenomenon. First, the intermolecular dipole-dipole interactions between OEG units, coupled with the hydrophobic methoxy (−OCH$_3$) end group of OEG, provide a favorable energy gain that drives the interdigitation of the OEG chains[20,41]. Second, the two phenolic units with OEG, connecting to the pentaerythritol core, offer a structural premise that facilitates interdigitation of the (3,4)-branching OEG chains (Supplementary Fig. 18c, d). This differs from previously reported interdigitation within a (3,5)-branching pattern[20]. The presence of partial interdigitation shifts the system out of equilibrium, leading to the formation of kinetically trapped MVVs. Upon increasing the temperature from R.T. to 37 °C, the weak interdigitation among OEG chains is disrupted, and the molecules revert to their more stable packing configuration, triggering the transition from MVVs to inverse cubosomes. Further increasing the temperature to 50 °C releases the conformational freedom embedded in the middle linking core, which restores the packing parameter to a favorable value for MLV formation (lamellar phase, $0.5 < p < 1$). This transition arises from the rotational freedom and the flip capability of molecules due to the carbon−carbon single bonds of the pentaerythritol unit (Fig. 1a)[20,42].

The rich energy landscape of self-assemblies from Janus dendrimer molecules with −OCH$_3$ end groups explicitly demonstrates the significant flexibility of these molecules, governed by packing parameters and dynamic non-covalent interactions (OEG interdigitation). The non-equilibrium MVVs were driven towards their equilibrium state through thermal annealing treatment. We hypothesized that we can

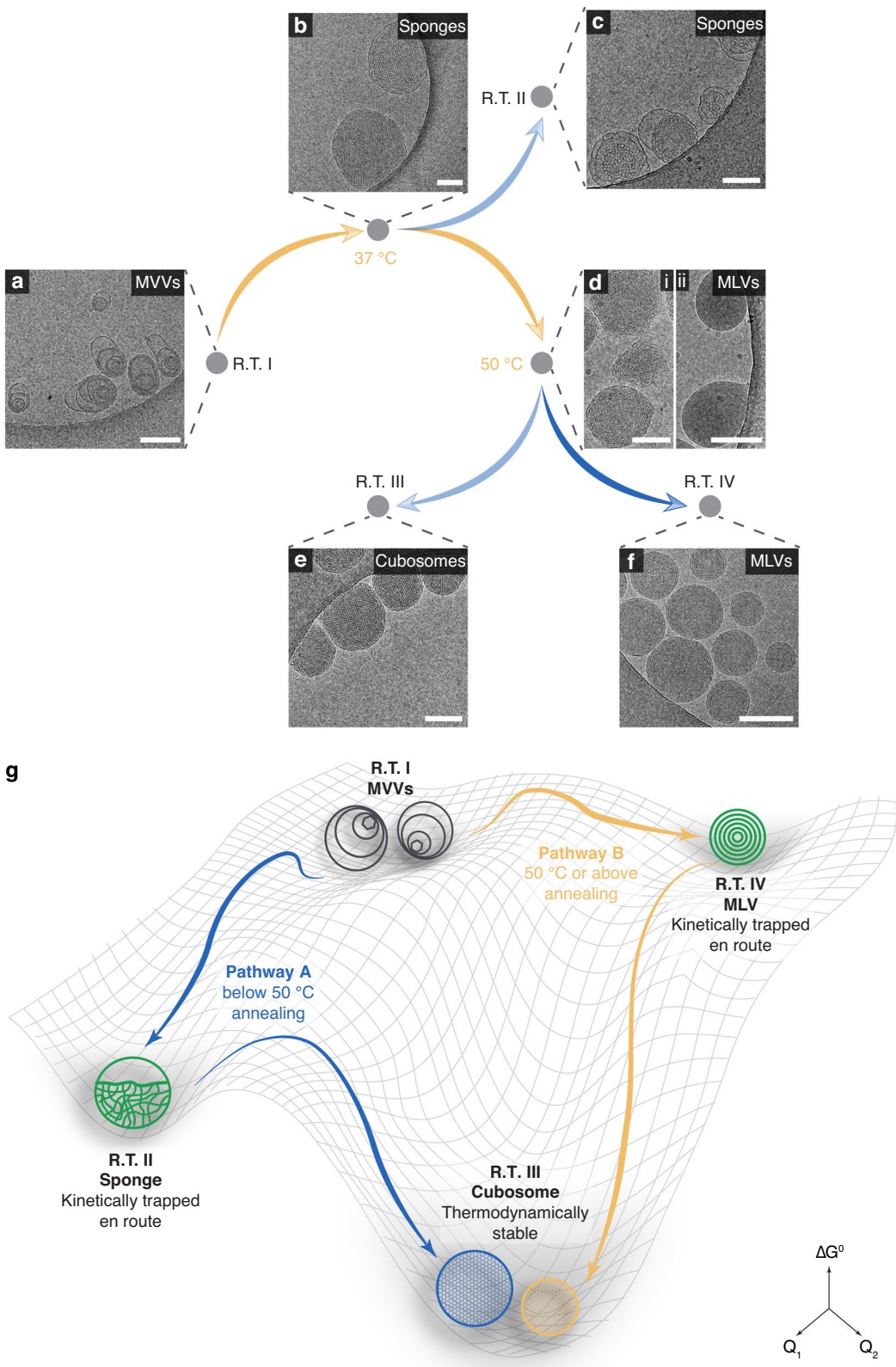

**Fig. 6 | Energy landscape and pathway selections of assemblies modulated by temperature. a–f** Cryo-TEM images of self-assemblies vitrified at the indicated temperatures during the heating/cooling cycle. Scale bars are 200 nm. **g** Schematic illustration of the pathway selections of assemblies at R.T., governed by the annealing temperatures. $Q_1$ and $Q_2$ denote coordinates for the lowest-energy transition pathway between the different self-assembly structures.

detain the system off route across the investigated temperature range by locking the interdigitation via hydrogen bonds within MVVs. Replacing the –OCH$_3$ end groups with hydroxy (–OH) groups introduces hydrogen bonding, which stabilizes OEG interdigitation up to 60 °C[20]. To substantiate this hypothesis, a type of Janus dendrimer with –OH end groups was synthesized (Supplementary Fig. 2), and the monodispersity of the synthesized molecules was confirmed by nuclear magnetic resonance (NMR) and matrix-assisted laser desorption/ionization time-of-flight (MALDI-TOF) (Supplementary Fig. 19). Upon performing self-assembly and dialysis using the same procedures, the newly self-assembled sample in the absence of ethanol displayed very similar size and PDI compared to the previous molecular system (Supplementary Fig. 20).

Furthermore, the replacement of –OCH$_3$ with –OH end groups did not pose significant morphological changes, and eccentric MVVs were clearly observed in the cryo-TEM images (Fig. 7a, R.T. I). Despite similar morphology, we anticipate that the interdigitation between bilayers within a single MVV is molecularly different, where hydrogen bonding exists among the –OH end groups. To test this, the energy landscape of the new self-assemblies was examined at different temperatures in real-time. After equilibrating at 37 °C, the original MVVs remained undisturbed without transforming into inverse structures (Fig. 7a, 37 °C). We attribute the enhanced stability of the vesicles to the locked interdigitation of the OEG chains via hydrogen bonding, which inhibits the molecular rearrangement (Fig. 7b). The locked interdigitation remained stable even at 50 °C, with no formation of inverse sponge intermediates or cubosomes observed (Fig. 7a, 50 °C). Interestingly, the MVVs exhibited a notable elongation transformation, resulting in the formation of tubular vesicles (TubeVs). The co-existence of ULVs indicates fission, which was found to occur in the non-interdigitated regions of TubeVs (highlighted by arrows in Fig. 7a, 50 °C). These findings further confirm the dynamic nature of these molecules, which drives transition to TubeVs and ULVs. Upon returning the system to R.T., MVVs resembling the original structures were recovered from the TubeVs (Fig. 7a, R.T. II). Through molecular design, we show that the self-assemblies can be guided toward energy landscapes, offering a pathway to control their dynamic behavior.

By leveraging the interplay between OEG interdigitation, conformational freedom, and molecular packing, we have extended the energy landscapes of self-assemblies into three-dimensional realms, enabling transitions between lamellar vesicles and inverse cubic structures. The ability of a single Janus dendrimer molecular system to drive transitions among energy states underscores its uniqueness. Distinct structures occupying different energy wells can be assessed simply by modulating thermal history, while the kinetics of the transitions and dimensions of inverse cubosomes can be precisely tuned through temperature and solvent conditions. To detain the system in out-of-equilibrium vesicular states, we encoded hydrogen bonding into the OEG corona, effectively locking bilayer interdigitation and revealing the underexplored potential and versatile chemistry of OEG (or its PEG counterpart)[20,43]. Our supramolecular assemblies span diverse structures, including vesicles, tubular vesicles, sponges, and inverse cubosomes, echoing the sophisticated structures naturally achieved by cells. Supported by our previous work[20], we expect that exploiting non-covalent interactions and dynamic molecular packing parameters to navigate rich energy landscapes of distinct morphologies provides a generalizable framework applicable to other classes of Janus dendrimers, including Janus glycodendrimers[44], ionizable amphiphilic Janus dendrimers[45], and stereochemical Janus dendrimer[46]. We further envision the rich landscapes of lamellar and inverse assemblies will advance the applications using Janus dendrimers for (m)RNA delivery[47–49], antimicrobial nanoreactors[50], and catalytic reactions within confined nanochannels[51].

## Methods
### Materials
All reagents are used as received without purification unless otherwise indicated. 1-bromododecane, pentaerythritol, and palladium on activated carbon (10% Pd, unreduced) were purchased from Acros Organics. Triethylene glycol monomethyl ether, triethylene glycol, *p*-

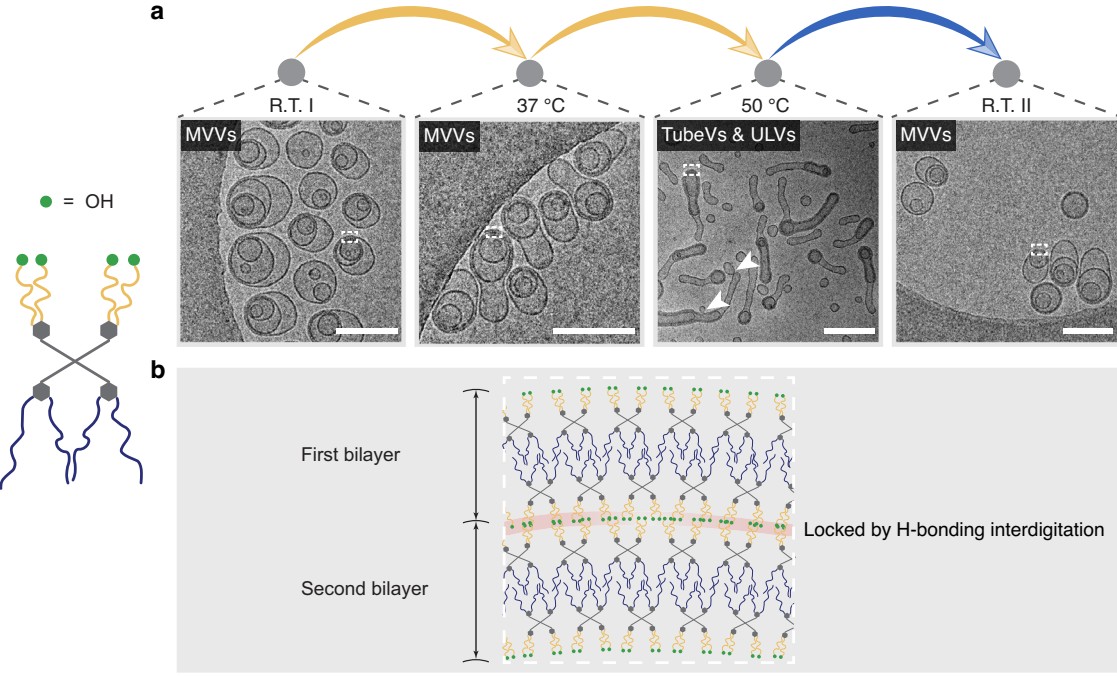

**Fig. 7 | Energy landscapes of self-assemblies from Janus dendrimers with –OH end groups in the absence of ethanol. a** Cryo-TEM images of self-assemblies vitrified at the indicated temperatures during the heating/cooling cycle. Arrows indicate non-interdigitated regions where membrane fission leads to the formation of ULVs. Scale bars are 200 nm. **b** Schematic illustration of locked interdigitation via hydrogen bonding between bilayers in selected regions highlighted by white dashed boxes in (**a**).

toluenesulfonyl chloride, methyl 3,5-dihydroxybenzoate, benzaldehyde, 4-(dimethylamino)pyridine, *p*-toluenesulfonic acid monohydrate, *N,N'*-dicyclohexylcarbodiimide, and benzyl bromide were products from Sigma-Aldrich. Dimethylformamide, potassium carbonate ($K_2CO_3$), sodium hydroxide (NaOH), sodium carbonate ($Na_2CO_3$), dichloromethane (DCM), and methanol were products from Thermo Fisher Scientific. Methyl 3,4-dihydroxybenzoate was purchased from TCI Europe NV. Ethanol (EtOH) and tetrahydrofuran (THF) were products from VWR International. Potassium hydroxide (KOH) was bought from J.T.Baker, Avantor. Hydrochloric acid (HCl, 37%) and potassium iodide (KI) were purchased from Merck. Dry THF and dry DCM were obtained by passing solvents over activated alumina columns in a MBraun MB SPS800 under nitrogen and stored under argon. Ultrapure Milli-Q water (QPOD Milli-Q purification system, 18.2 MΩ) was used for the preparation of all non-deuterated aqueous solutions.

### Synthesis and characterization of Janus dendrimers
All Janus dendrimers were synthesized following previously reported protocols, with their purity confirmed by NMR and MALDI-TOF analysis. Detailed description can be found in the **Supplementary Information**.

### General procedure for self-assembly preparation
Self-assembly was performed by rapidly injecting (approximately 0.5 s) 100 μL of dendrimer solution in absolute ethanol into 2 mL of Milli-Q water, followed by 5 s of vortex mixing. The final dendrimer concentration is approximately 0.5 mg/mL. To remove ethanol from the prepared samples, the solution was transferred into Spectra/Por® dialysis tubing (molecular weight cut-off: 3.5 kDa) and dialyzed against a large volume of Milli-Q water with frequent medium changes under continuous stirring. The residual ethanol concentration was below 4.8 ppm as determined by $^1$H NMR.

### Cryogenic transmission electron microscopy (Cryo-TEM)
**Cryo-TEM imaging at room temperature.** Cryo-TEM imaging was conducted with a JEOL 2100 Transmission Electron Microscope operating at 200 kV, equipped with a high-resolution Gatan 895 Ultrascan 4000 bottom-mount camera (4080 × 4080 pixels) to capture the morphologies of self-assemblies. TEM grids (Quantifoil®) were glow-discharged by a 208 carbon coater (Cressington). To ensure sufficient sample concentration, the original sample solution was concentrated via centrifugation. Subsequently, 3.5 μL of the concentrated sample was loaded onto the grid, blotted, and vitrified by plunging into liquid ethane at 100% humidity using an FEI Vitrobot™ Mark IV (blot time: 1.5 s, blot force: 2). The samples were loaded into a Gatan 914 High tilt cryoholder (Munich, Germany) and inserted into the microscope for imaging. Data analysis was performed using Fiji imageJ (v 2.1.0).

**Cryo-TEM imaging at elevated temperatures.** Cryo-TEM imaging above room temperature (37, 41, 44, 50, and 60 °C) was performed to investigate sample morphologies during the heating/cooling cycles. The Vitrobot chamber was set to the target temperature and 100% relative humidity. Elevated temperatures introduced challenges due to water vapor condensation, which hindered sample preparation with optimum ice thickness formation for imaging. To address these issues, a custom-made device constructed from a Falcon tube was implemented to direct water vapor flow. Additionally, minimizing the clamping contact area during vitrification significantly enhanced the success rate for preparing grids with thin ice. Heating/cooling cycles were carried out using an Apollo Thermal Cycler (Model ATC401) with programmed temperature ramping. Samples were equilibrated at each temperature for 10 min before vitrification. To prevent any temperature drop affecting the sample, Eppendorf tips were pre-heated to the target temperatures on a hot plate. Tweezers and grids were incubated in the Vitrobot chamber for 2 min at the respective temperatures

before sample preparation. The process involved loading 6 μL of sample onto the grid, resting for 2 s, blotting (blot time: 1.5 s, blot force: 0), and vitrification in liquid ethane. Prepared samples were transferred to the cryo-holder and imaged using the same microscope settings as described previously.

**Quantification and characterization of self-assemblies.** Cryo-TEM was used to examine the samples, and the percentage of different structures was quantified for each sample. To ensure an accurate representation of the sample's state, images were randomly collected from multiple grid regions and batches, with at least 500 particles counted in total for each condition. The results were presented as bar charts, depicting the percentage frequency of each self-assembly morphology. Regions of interest were subjected to the fast Fourier transform (FFT) using Fiji imageJ (v 2.1.0).

### Reporting summary
Further information on research design is available in the Nature Portfolio Reporting Summary linked to this article.

## Data availability
Data supporting the findings of this study are available within the Article and its Supplementary Information. The source data underlying Figs. 2–5 and Supplementary Figs. 4, 7, 8, 10, 11, 15, 17, and 20 are provided as a Source Data file. Source data are provided with this paper.

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

## Acknowledgements

This project has received funding from the Ministry of Education, Culture and Science (Gravitation program 024.001.035) (to D.A.W.) and from the European Research Council (ERC) under the European Union's Horizon 2020 research and innovation programme ERC-CoG 101044434 "SynMoBio" (to D.A.W.). J.L. acknowledges Dr. Shuquan Cui for the helpful discussion. I.K.V. acknowledges the financial support of the European Research Council (ERC-2020-CoG 101001965) and the Dutch Ministry of Education, Culture and Science (Gravity program 024.005.020). N.M. is grateful to Dr. Bence Fehér for providing valuable feedback on the analysis of SAXS profiles.

## Author contributions

J.L. and D.A.W. conceived the concept. J.L. designed and performed most of the experiments, analyzed the results, and prepared the

manuscript. D.W. assisted in the kinetic study of self-assemblies and manuscript preparation. N.P.K. contributed to the synthesis and characterization of Janus dendrimers. N.M. performed the SAXS measurements. N.M. and I.K.V. analyzed SAXS experiments. All authors discussed the results and commented on the manuscript.

## Competing interests

The authors declare no competing interests.
