## [Transparent Peer Review file · Nature Communications]

Dynamic pathways in energy landscapes guiding supramolecular Janus dendrimer self-assemblies between lamellar and cubic architectures

Corresponding Author: Professor Daniela Wilson

Version 0:

Reviewer comments:

Reviewer #1

(Remarks to the Author)

This manuscript builds on the authors' previous report on the complex energy landscape of self-assembled vesicles [J. Am. Chem. Soc. 145, 15496-15506 (2023)] by investigating how a related Janus dendrimer-bearing inverted positions of the aliphatic and oligo(ethylene glycol) units-influences the self-assembly pathways. The finding that lamellar vesicles and inverted cubic structures can reversibly interconvert is interesting, especially as a time-evolution phenomenon in an artificial self-assembly system. However, the overall discussion remains largely qualitative: no quantitative estimation of free energy or activation barriers for the various assembled states is presented, leaving the energy landscape descriptions of the paper somewhat speculative. Since many recent studies in supramolecular chemistry provide more quantitative characterization of thermodynamic stability and kinetic parameters, it is not entirely clear how this work substantially advances these discussions. Nevertheless, the new molecular design may provide insights into inverse cubic phases that were not observed in the authors' previous study. I encourage the authors to emphasize how these structural differences lead to novel assembly phenomena. The time evolution of the supramolecular assemblies is intriguing, but the suitability of the manuscript for publication in Nature Communications warrants re-evaluation after major revisions.

Comments

1. Residual Ethanol Evaluation

As in the authors' previous work, the presence or absence of ethanol significantly affects the resulting amphiphilic dendrimer assemblies. The Methods section indicates ethanol removal by dialysis, but the paper provides no experimental data on the residual ethanol levels afterward. Given that even trace amounts of water/solvent can alter supramolecular assembly (see: Zee et al., Nature 558, 100–103 (2018); Rao et al., Nat. Chem. 9, 1133–1139 (2017)), it is essential to confirm the residual ethanol concentration (e.g., by GC or HPLC) to validate the comparisons made in the manuscript.

2. Reliability of the Energy Landscapes

Figures 2 and 6 propose energy landscapes for assembly formation, yet there is no direct evaluation of the relative thermodynamic stabilities or pathways of each morphological state. Consequently, the energy diagrams appear hypothetical rather than empirically grounded. For instance, the axes "Q1" and "Q2" in Figure 6g remain undefined. If pathways 2 and 3 occur at the same temperature, why does pathway 1 involve a different temperature? Clarifying these points would strengthen the paper's claims about the energy landscape.

3. Concentration Dependence and Pathway Analysis

The concepts of "on-pathway" and "off-pathway" in self-assembly often require examining concentration effects. This manuscript focuses on one concentration, so I recommend at least a brief discussion of how different concentrations might alter the assembly pathways—or, if a single concentration is critical, explaining why it was chosen. This would give a clearer picture of the system's kinetics and thermodynamics.

4. Thermodynamic Stability Measurements

The morphological transitions from MVV (multivesicular vesicles) to sponges and then to cubosomes are discussed only in a conceptual manner. Can calorimetric (e.g., DSC) or spectroscopic (e.g., IR) methods be used to assess the relative stability of each morphological state? Such data would help confirm whether these phase transitions stem from meaningful

thermodynamic differences versus purely kinetic traps.

5. Rationale for the 37 °C Experiments

The manuscript describes several experiments at 37 °C, possibly reflecting an interest in physiological conditions. If that is indeed the motivation, it should be explicitly stated, as it would contextualize the choice of temperature.

6. Bio-Application Claims

While the introduction and conclusion mention potential biomedical applications of these dendrimer assemblies, the system's high sensitivity to ethanol (and possibly other additives) suggests it may not yet be straightforward to translate into complex biological environments. If bio-applications are a genuine aim, demonstration of assembly formation in relevant buffer solutions would be much more convincing. If not, I don't think the authors need to describe anything related to bio-applications.

Reviewer #2

(Remarks to the Author)

The study explores Janus dendrimer self-assemblies, which can transition reversibly between lamellar vesicles and inverse cubic structures, featuring a complex energy landscape driven by temperature-triggered non-covalent interactions. These assemblies are characterized by a variety of structures, including vesicles, tubular vesicles, sponges, and cubosomes. The energy landscapes are modulated by molecular packing, interdigitation of OEG chains, and hydrogen bonding, with temperature acting as a key factor in shifting between different energy states and affecting packing parameters and assembly pathways. The transition from lamellar to inverse cubic structures is achieved by controlling the packing parameter. The Janus dendrimers provide a versatile system with applications in biomedicine, catalysis, and nanoreactor design. The study highlights the dynamic nature of self-assemblies, which can be kinetically trapped into metastable states. A deeper understanding of these molecular interactions may aid in advancing self-assembly techniques for diverse applications, and the findings propose that these dynamic systems could mimic complex cellular behaviors, offering new insights into supramolecular chemistry. The cryo TEM images are impressive, however, certain aspects of the manuscript needs clarification and supporting data. Thus, I recommend major revision of the manuscript after addressing the following points.

Comments:

1. What is the significance of the annealing temperature of 37 °C and 50 °C in this context, considering that oligo(ethylene glycol) (OEG) units exhibit lower critical solution temperature (LCST) behavior? What is the LCST of the present molecules? Is the annealing temperature above or below the LCST of OEG, and how does this influence the self-assembly process?
2. LCST is crucial to say further about the kinetic states. The authors may consider recording temp dependent UV-vis spectra to follow turbidity. Additionally, the studies were conducted at temperatures of 37 °C and 50 °C. What is about performing particle size analysis using Light Scattering (LS), Nanoparticle Tracking Analysis (NTA), and Cryo-TEM at RT as it was anyway annealed to RT.
3. Also, such kinetic state formation also depends on the rate of annealing. However, I could not find such different annealing kinetics to compare the states. It is crucial to investigate the impact of the cooling rate on the morphological transition, as only two cooling conditions (6 °C/min and rapid quenching using ice bath or liquid nitrogen) have been considered. Therefore, it is recommended to cool the system at a slower rate, such as 1 °C/min, to assess the effect of a more gradual cooling process on the transition dynamics.
4. Based on the temperature, the polymeric nanostructures varies. However, how that correlates to the packing parameter was not clear. No discussion was attempted as well.
5. Why the control molecule without the methoxy was taken. The additional hydrogen bonding might change the kinetic states, energetics and lead to different LCST behaviors. The relevant discussion was also condensed in a small paragraph. Detailed experiments with the control molecules just like the main molecules might show the differences.
6. SAXS data in the ESI was qualitative. Can it be used to distinguish between the polymeric structures or the ordering?
7. In line 126, figure number of supplementary information is wrong.
8. ESI: Page 3, Scheme 1, molecular structures of 8 is to be corrected.

Reviewer #3

(Remarks to the Author)

The manuscript by Luan and co-workers presents an investigation into kinetically controlled supramolecular systems, utilizing temperature-driven transitions to navigate complex energy landscapes. By employing lamellar vesicles and inverse cubic structures, the authors demonstrate how amphiphilic moieties capable of hydrogen bonding enable dynamic pathway selection. A particularly striking aspect of this study is the clear demonstration of thermal history dependence, where the structural transitions are dictated by the applied temperature and solvent conditions. The authors employ a different set of microscopic and spectroscopic techniques to characterize their systems, offering important insights into how molecular packing, flexibility, and non-equilibrium self-assembly govern structural evolution. Their detailed discussion on the interplay between packing parameters and reconfigurable energy landscapes provides a compelling framework for understanding the

design rules of supramolecular architectures. This work effectively bridges supramolecular chemistry, non-equilibrium control, and adaptive self-assembly, with promising implications for biological applications. Moreover, it opens alternative directions in the study of the origins of life, where transient assemblies could have emerged and persisted under temperature gradients, forming dynamic structures with defined lifetimes. Overall, the manuscript represents an important contribution to the field. I recommend publication in Nature Communications, pending the requested revisions to clarify specific mechanistic aspects and improve readability in certain sections.

1) The study centers around temperature-driven assembly pathways, with detailed comparisons between 37 °C and 50 °C. Could the authors clarify the rationale behind the selection of these specific temperatures? Additionally, it would be informative to understand whether the system displays any notable behavior in the intermediate temperature range, as this could provide further insight into the transition kinetics and possible intermediate states.

2) The manuscript highlights an interesting interplay between temperature and annealing time in directing distinct assembly pathways. While the difference in structural outcomes at 37 °C (72 h) and 50 °C (10 min) is striking, it remains unclear whether time alone at intermediate temperatures could yield similar transitions. Could the authors elaborate on whether prolonged annealing at lower temperatures might eventually access the same cubosome phase as rapid heating to 50 °C? Conversely, is there a critical temperature–time threshold beyond which pathway selection becomes irreversible or biased?

3) Given the elegant demonstration of structural transitions driven by temperature and thermal history, have the authors considered using confocal or super-resolution microscopy techniques to capture these dynamics in real time? While the final structures may be below the resolution limit for direct observation, such approaches could potentially offer valuable insight into fusion events, morphological transitions, or heterogeneity within the population—especially during early stages of annealing. Even partial visualization might complement the cryo-TEM and scattering data, offering a richer picture of the system's kinetic behavior.

4) The manuscript compellingly shows how both annealing temperature and solvent influence transition kinetics and particle dimensions, particularly highlighting ethanol's role in enhancing cubosome growth via apparent fusion processes. While one might be tempted to ask about changing the solvent system altogether, this is not the intent here. Rather, could the authors elaborate on how ethanol subtly reshapes the energy landscape—possibly by modulating hydration, packing parameters, or membrane fluidity? Is there a concentration threshold beyond which fusion becomes strongly favored? A more detailed discussion of how solvent environment and thermal history co-regulate the transition pathways would enrich the mechanistic understanding of this finely tuned supramolecular system.

5) The manuscript presents a nice demonstration of how different thermal pathways give rise to multiple distinct self-assembled states, including MVVs, sponges, MLVs, and cubosomes. Given the system's sensitivity to thermal history and cooling rates, it would be interesting to understand its robustness across repeated thermal cycles. Can the authors comment on how many heating/cooling cycles the system can undergo before irreversible changes—such as degradation, kinetic trapping, or loss of responsiveness—begin to occur?

6) Given the system's sensitivity to thermal history and the presence of kinetically trapped states, could the authors comment on whether the age of the sample prior to annealing affects the subsequent assembly behavior? For instance, would allowing the sample to rest at room temperature for extended periods (e.g., hours to days) before initiating thermal treatment influence the accessibility or stability of certain morphologies? Exploring whether pre-annealing aging introduces additional kinetic constraints—or perhaps even facilitates alternative pathways—would provide deeper insight into the temporal plasticity of these supramolecular assemblies.

7) The redesign with –OH end groups is a strong demonstration of how molecular interactions can reshape the energy landscape. It would be helpful if the authors clearly stated which parameters—such as solvent conditions, presence of ethanol, or other additives—were used in these experiments. For the –OH terminated dendrimers, was any co-solvent involved? Moreover, could the authors comment on the generality of these findings? Are the observed transitions and morphologies specific to this Janus dendrimer design, or might these principles extend to related systems?

Version 1:

Reviewer comments:

Reviewer #1

(Remarks to the Author)

I have carefully reviewed the revised manuscript. The authors addressed the reviewers' comments exceptionally thoroughly and sincerely, adding numerous experimental results and providing credible responses to all the concerns raised. I wish to express my sincere commendation to the authors for their diligent efforts. The manuscript aligns well with the scope of Nature Communications, and I strongly recommend publishing it in its current form.

Reviewer #2

(Remarks to the Author)

I have gone through the authors' responses not only to my comments but also the concerns raised by other reviewers as

well. In my view the manuscript has now improved significantly upon incorporating the suggested experiments and quantitative analysis. Further, manuscript has also become easy to follow upon incorporating the schemes.

Thus, in my view manuscript may now be accepted in Nature Comm upon incorporating the following minor aspects:

1. Page 3, Scheme 1, molecular structures of 8 has few bonds missing. Please check if they go missing during PDF conversion.
2. The author may cite the following recent paper that talks about packing parameter mediated transformation of vesicle to micelles. ChemComm, 2023, 59, 13195 – 13198.

Reviewer #3

(Remarks to the Author)

The authors have addressed all of my comments and concerns in their revision. I recommend acceptance of the manuscript in its current form.

Response Letter

We would like to express our sincere gratitude to the reviewers for their valuable comments and suggestions on our manuscript, which have helped us to considerably improve our revised manuscript. In addition to the point-to-point response to the reviewers, we have performed additional experiments and provided additional information to strengthen and support the relevance of our findings.

A point-by-point response to the reviewers' comments is provided below, where reviewers' comments are marked in black while **our response is in blue**. The revised Manuscript with **changes highlighted in red** has been provided.

Reviewers comments:

Reviewer #1:

This manuscript builds on the authors' previous report on the complex energy landscape of self-assembled vesicles [J. Am. Chem. Soc. 145, 15496-15506 (2023)] by investigating how a related Janus dendrimer-bearing inverted positions of the aliphatic and oligo(ethylene glycol) units-influences the self-assembly pathways. The finding that lamellar vesicles and inverted cubic structures can reversibly interconvert is interesting, especially as a time-evolution phenomenon in an artificial self-assembly system. However, the overall discussion remains largely qualitative: no quantitative estimation of free energy or activation barriers for the various assembled states is presented, leaving the energy landscape descriptions of the paper somewhat speculative. Since many recent studies in supramolecular chemistry provide more quantitative characterization of thermodynamic stability and kinetic parameters, it is not entirely clear how this work substantially advances these discussions. Nevertheless, the new molecular design may provide insights into inverse cubic phases that were not observed in the authors' previous study. I encourage the authors to emphasize how these structural differences lead to novel assembly phenomena. The time evolution of the supramolecular assemblies is intriguing, but the suitability of the manuscript for publication in Nature Communications warrants re-evaluation after major revisions.

We would like to thank the reviewer for the positive remarks. We have performed additional experiments to address all the insightful comments and incorporated helpful suggestions into our revised manuscript.

“quantitative estimation of free energy or activation barriers”: To obtain the energy barrier of the transition from original MVVs to cubosomes in the absence of ethanol, we quantitatively analyzed the transition by cryo-TEM in the temperature range from 37 to 44 °C. The reason for choosing this temperature range is because the transition follows the same pathway (Pathway 1, updated **Fig. 6g**). Please see **Point 2** for detailed explanation. The transition was found to follow a second-order kinetics. By means of an Arrhenius plot, the energy barrier for the transition was calculated to be 109 kJ mol⁻¹ (**Fig. 1 to Reviewer #1**).

Fig. 1 to Reviewer #1 | Energy barrier of the transition from kinetically trapped vesicles to the thermodynamic cubosomes. **a** Quantitative analysis of assembly populations in the absence of ethanol annealing for different time periods at different temperatures. Images from multiple areas and batches were analyzed to minimize the error (Particles per condition: $n > 500$ for 37 °C annealing, $n > 800$ for 41 °C annealing, and $n > 1100$ for 44 °C annealing). **b** Second-order kinetic plot at specific annealing temperatures against time for the transformation of MVVs to cubosomes. **c** An Arrhenius plot of the transition from MVVs to cubosomes. The K values were obtained by fitting the population of cubosomes with a second-order rate equation in **(b)**. **d** Schematic representation of the energy landscape of self-assemblies of MVV and Cubosome separated by an energy barrier (E_b) of 109 kJ mol⁻¹.

“I encourage the authors to emphasize how these structural differences lead to novel assembly phenomena.”: Compared with recent studies in supramolecular systems of mostly one-dimensional fibers, our study advances supramolecular landscapes in three-dimensional realms, which are widely prevalent in cellular structures. Using single-molecular self-assembly system, a rich energy landscape with reversible transitions between lamellar (kinetically trapped) and inverse cubic (thermodynamic) structures. Through the interplay of oligo(ethylene glycol) (OEG) interdigitation, conformational flexibility, and molecular packing, we successfully navigated the assemblies through distinct thermodynamic wells, demonstrating precise control over their structural transitions.

“To further quantify the energy barrier of the MVV-to-cubosome in **Pathway 1**, we performed cryo-TEM analysis across the temperature range of 37 °C to 44 °C. By means of an Arrhenius plot, the energy barrier for the transition was calculated to be 109 kJ mol⁻¹ (**Supplementary Fig. 12**).” and related discussion were added to the revised **Manuscript**.

Fig. 1 to Reviewer #1 and related description were added to **Supplementary Information**.

Comments

1. Residual Ethanol Evaluation

As in the authors' previous work, the presence or absence of ethanol significantly affects the resulting amphiphilic dendrimer assemblies. The Methods section indicates ethanol removal by dialysis, but the paper provides no experimental data on the residual ethanol levels afterward. Given that even trace amounts of water/solvent can alter supramolecular assembly (see: Zee et al., Nature 558, 100–103 (2018); Rao et al., Nat. Chem. 9, 1133–1139 (2017)), it is essential to confirm the residual ethanol concentration (e.g., by GC or HPLC) to validate the comparisons made in the manuscript.

Reviewer 1, Answer point 1: Thank the reviewer for this important comment. To confirm the residual ethanol concentration after dialysis, we employed ¹H NMR to examine the potential presence of ethanol in the self-assembly samples.

Fig. 2 to Reviewer #1 | Residual ethanol determination by NMR. Ethanol solution of 4.8 ppm (mass concentration) and self-assembly samples from three different batches after dialysis were tested using D₂O as the solvent with 500 MHz NMR.

Self-assembly samples from three different batches after dialysis together with a control ethanol sample (4.8 ppm) were scanned by 500 MHz NMR with high sensitivity (**Fig. 2 to Reviewer #1**). While the 4.8 ppm ethanol sample displayed clear proton peaks belonging to $\text{CH}_3\text{CH}_2\text{OH}$, no detectable ethanol peaks could be found in the spectra of self-assembly samples. Therefore, we could conclude that the residual ethanol concentration is far below 4.8 ppm after the through dialysis.

This information was added to the revised **Manuscript**.

2. Reliability of the Energy Landscapes

Figures 2 and 6 propose energy landscapes for assembly formation, yet there is no direct evaluation of the relative thermodynamic stabilities or pathways of each morphological state. Consequently, the energy diagrams appear hypothetical rather than empirically grounded. For instance, the axes “Q1” and “Q2” in Figure 6g remain undefined. If pathways 2 and 3 occur at the same temperature, why does pathway 1 involve a different temperature? Clarifying these points would strengthen the paper’s claims about the energy landscape.

Reviewer 1, Answer point 2: Thank the reviewer for the valuable comment, allowing us to improve the Figure to clarify our points.

We apologize for missing the information of “ Q_1 , Q_2 ”. Q_1 and Q_2 denote coordinates for the lowest-energy transition pathway between the different self-assembly structures^{1,2}. This has been added to the caption of **Fig. 6g** in the revised **Manuscript**.

The thermodynamic stabilities of each morphology during the transition from MVVs to cubosomes will be discussed with additional experimental results in **Point 4** as asked by the reviewer.

The pathway transitions among different morphological states were reliable and their transition kinetics were carefully and systematically characterized by LS, NTA, and cryo-TEM at different annealing temperatures (37 °C, **Fig. 2**; 44 °C, **Fig. 2 to Reviewer #3**; 50 °C, **Fig. 3**). The effect of the presence of ethanol solvent on the morphological transition kinetics was also investigated and compared (**Fig. 4** and **Fig. 5**). By checking the structures during the heating and cooling process during the annealing (**Fig. 6a–f**), we identified the different pathways of the transition from MVVs to cubosomes. With all these results, we summarized our findings in a schematic energy landscape of the current self-assembly system in **Fig. 6g**. Therefore, we respectfully disagree that “our energy diagrams appear hypothetical”. But we noticed that the previous version of **Fig. 6g** was not presented in a clear way, which caused some misunderstandings. Together with additional experiments as suggested by reviewers, we have updated this illustration as follows:

Updated **Fig. 6g** | Schematic illustration of the pathway selections of assemblies at R.T., governed by the annealing temperatures. Q_1 and Q_2 denote coordinates for the lowest-energy transition pathway between the different self-assembly structures.

This illustration depicts the energy landscape of assemblies at R.T., including three kinetically trapped states and one thermodynamically stable state. We would like to first guide the reviewer's attention only to the starting state (MVV, R.T. I) and the thermodynamically stable state (Cubosome, R.T. III). After annealing the self-assembly solutions (heating to a higher temperature for certain periods of time, then cooling down to R.T.), the original MVVs transformed to cubosomes. Such morphological transitions, as demonstrated by cryo-TEM imaging (Fig. 2d–k and Fig. 3d–g), were accompanied with noticeable PDI and count rate changes in LS (Fig. 2a and Fig. 3a). Altering the cooling rates (1 °C/min, 6 °C/min, or room-temperature air quenching) did not pose any effect on the transition to cubosomes (Supplementary Fig. 13a–c). Furthermore, the size, PDI, count rates (Supplementary Fig. 2) and the morphology (Supplementary Fig. 6) of the as-formed cubosomes remained constant in the second annealing cycle. All the above observations reveal that the cubosomes are the thermodynamically favored product, residing in the lowest energy well of the energy landscape at R.T. This process could be summarized in Fig. 3 to Reviewer #1.

Fig. 3 to Reviewer #1 | Schematic representation of the energy landscape of self-assemblies of MVV and Cubosome.

Next, we would like to discuss the pathway selections and the two kinetically trapped states (Sponge, R.T. II and MLV, R.T. IV) en route to the thermodynamically favored cubosomes. Although annealing at 37 °C and 50 °C could both bring the MVVs to cubosomes, the much faster transition kinetics at 50 °C annealing (10 min) than 37 °C annealing (72 h) was attributed to the high energy input. The different sizes of the final cubosomes seemed to indicate annealing at these two temperatures might follow different pathways (Fig. 5a,b). To elucidate this, we closely monitored the morphologies of assemblies with cryo-TEM in real-time during the annealing process (Fig. 6a–f). We found there existed two pathways for the MVVs to transform to cubosomes:

Pathway 1, 37 °C annealing: MVVs (lamellar phase, at R.T.) → sponges/cubosomes (inverse phase, at 37 °C and R.T.);

Pathway 2, 50 °C annealing: MVVs (lamellar phase, at R.T.) → sponges/cubosomes (inverse phase, at 37 °C) → MLVs (lamellar phase, at 50 °C) → cubosomes (inverse phase, at R.T.).

The main difference lies in the remarkable formation of MLVs at 50 °C, which are a vesicular type of lamellar phase. The unexpected transition from the inverse phases at 37 °C back to lamellar MLVs showed a different pathway of formation of cubosomes than that observed at 37 °C annealing.

To identify the pathway selection temperature, we chose two additional temperatures, below and above 50 °C, to monitor the real-time transitions throughout the annealing process. For the temperature below 50 °C, an intermediate temperature between 37 °C and 50 °C of 44 °C was chosen. For the temperature above 50 °C, we chose 60 °C, which is the highest temperature that could be obtained with the Vitrobot vitrification system for cryo-TEM sample preparation.

Fig. 4 to Reviewer #1 | Pathway selections of self-assemblies via 44 °C and 60 °C annealing. a–d Cryo-TEM images of self-assemblies vitrified at the indicated temperatures during the

heating/cooling cycle. Self-assemblies were vitrified after being equilibrated at 44 °C and 60 °C for 35 min and 10 min, respectively. Scale bars are 200 nm.

Annealing self-assemblies at 44 °C followed the same pathway with 37 °C annealing (Pathway 1, **Fig. 6b**) as inverse sponges/cubosomes were found at 44 °C (**Fig. 4a,b to Reviewer #1**). When the self-assemblies were annealed at 60 °C, the transition followed Pathway 2 with the formation of the lamellar MLVs structures at 60 °C (**Fig. 4c to Reviewer #1**). Combining all these results, we can identify a critical pathway selection temperature of 50 °C of the transitions from MVVs to cubosomes.

After unveiling the two pathway selections, we sought to map the energy landscape of self-assemblies at R.T. with distinct morphologies by kinetically trapping the structures en route to thermodynamically stable cubosomes. Sponges, which act as precursors to cubosomes, were readily trapped by interrupting the annealing process at 37 °C, owing to the slow kinetics of the transition (**Fig. 2k**). As a kinetic intermediate, the Sponge state (R.T. II, **Fig. 6g**) occupies an energy minimum between the initial MVVs (R.T. I, **Fig. 6g**) and final Cubosome state (R.T. III, **Fig. 6g**). In contrast, MLVs only emerged above 50 °C suggesting a higher energy cost, as reflected by the difficulty in kinetically trapping these structures at R.T. (**Supplementary Fig. 13**). The transition to cubosomes occurred consistently at R.T. upon cooling from 50 °C via different cooling rates from 1 °C/min to rapid 0 °C quenching (**Supplementary Fig. 13**). Only through an ultrafast quenching procedure using liquid nitrogen could successfully trap lamellar MLVs at R.T., resulting in a distinct fourth energy state (R.T. IV, **Fig. 6f,g** and **Supplementary Fig. 13d**). Notably, ultrafast quenching method enabled a transition from thermodynamic cubosomes back to kinetically trapped vesicular MLV structures. All four states remained stable over several days at R.T., indicating that each resides in a local energy minimum (please see **Point 4** and **Supplementary Fig. 13e**).

All the above results were schematically represented in the updated **Fig. 6g** in the revised **Manuscript** with updated discussions:

“All the above findings demonstrate that at R.T., the energy-favored packing parameter of the molecules exceeds unity, driving the transition from original lamellar MVVs (R.T. I) to cubosomes (R.T. III). Interestingly, annealing at 37 °C and 50 °C led to two distinct transformation pathways from MVVs to cubosomes. In **Pathway 1**, annealing MVVs (lamellar phase, at R.T.) at 37 °C resulted in a gradual transformation into sponges and subsequently cubosomes (inverse phases, at both 37 °C and R.T.). In contrast, in **Pathway 2**, annealing at 50 °C produced MLVs (lamellar phase, at 50 °C) from the inverse structures, which then converted into cubosomes upon cooling to R.T.. The key distinction between these pathways lies in the formation of MLVs at 50 °C, a vesicular lamellar phase not observed at 37 °C. To further define the temperature threshold for pathway selection, we examined two annealing temperatures, 44 °C and 60 °C, below and above 50 °C, respectively, to monitor the real-time transitions throughout the annealing process. Annealing at 44 °C followed **Pathway 1** with

inverse sponges and cubosomes observed (Supplementary Fig. 11a,b), whereas annealing at 60 °C followed Pathway 2, featuring MLV formation at 60 °C and cubosomes upon returning to R.T. (Supplementary Fig. 11c,d). Together, these results identify 50 °C as a critical transition temperature that governs the pathway selection from MVVs to cubosomes. To further quantify the energy barrier of the MVV-to-cubosome in Pathway 1, we performed cryo-TEM analysis across the temperature range of 37 °C to 44 °C. By means of an Arrhenius plot, the energy barrier for the transition was calculated to be 109 kJ mol⁻¹ (Supplementary Fig. 12).”

“We next sought to map the energy landscape of self-assemblies at R.T. with distinct morphologies by kinetically trapping the structures en route to the thermodynamically stable cubosomes. Sponges, which act as precursors to cubosomes, were readily trapped by interrupting the annealing process at 37 °C, owing to the slow kinetics of the transition (Fig. 2k). As a kinetic intermediate, the Sponge state (R.T. II, Fig. 6g) occupies an energy minimum between the initial MVVs (R.T. I, Fig. 6g) and final Cubosome state (R.T. III, Fig. 6g). In contrast, MLVs only emerged above 50 °C suggesting a higher energy cost, as reflected by the difficulty in kinetically trapping these structures at R.T. (Supplementary Fig. 13).”

Fig. 1b was also updated accordingly.

Fig. 4 to Reviewer #1 was added to Supplementary Information.

3. Concentration Dependence and Pathway Analysis

The concepts of “on-pathway” and “off-pathway” in self-assembly often require examining concentration effects. This manuscript focuses on one concentration, so I recommend at least a brief discussion of how different concentrations might alter the assembly pathways—or, if a single concentration is critical, explaining why it was chosen. This would give a clearer picture of the system’s kinetics and thermodynamics.

Reviewer 1, Answer point 3: We thank the reviewer for this helpful comment. The current self-assembly process follows the injection of 100 µL of dendrimer solution in ethanol (10 mg/mL) into 2 mL of Milli-Q water, resulting in a concentration of approximately of 0.5 mg/mL (to be accurate, 0.48 mg/mL). We chose this protocol to ensure the reproducibility to acquire self-assemblies with monomodal distributions and a low polydispersity (PDI < 0.15). Increasing or decreasing the concentrations could lead to assemblies with higher PDI or even bimodal distributions, which was reported previously^{3,4}.

To further check whether there is a concentration effect on the pathways of the self-assembly system, we prepared self-assemblies of two more concentrations, one higher (1 mg/mL) and one lower (0.1 mg/mL) and checked if they followed the same morphological transition pathway after the annealing treatment at 50 °C.

The self-assembly samples were prepared using a similar protocol as follows:

Sample 0.1 mg/mL: injection of 100 µL of dendrimer solution in ethanol (2 mg/mL) into 2 mL of Milli-Q water;

Sample 1 mg/mL: injection of 100 μ L of dendrimer solution in ethanol (20 mg/mL) into 2 mL of Milli-Q water.

After thorough dialysis to remove the ethanol, both samples showed a monomodal distribution with much higher PDI (0.1 mg/mL, PDI 0.30; 1 mg/mL, PDI 0.24; **Fig. 5a,b to Reviewer #1**) than the self-assembly from concentration of 0.5 mg/mL in the manuscript (PDI 0.15). Regardless to the difference in concentrations, similar nonconcentric multivesicular vesicles (MVVs) were found to be the dominant population of both self-assemblies (**Fig. 5c,e to Reviewer #1**). Upon annealing at 50 °C for 10 min, MVVs were replaced by inverse cubosomes (**Fig. 5d,f to Reviewer #1**). These results confirm that the investigated concentrations of self-assemblies (0.1 – 1 mg/mL) do not pose a significant effect on the transitions of assembly pathways.

Fig. 5 to Reviewer #1 | Investigation on the transitions of self-assembly pathway at different concentrations. a,b D_h profiles of self-assemblies at concentrations of 0.1 mg/mL (a) and 1 mg/mL (b) determined by LS. Ethanol was removed by thorough dialysis. D_h and PDI of the self-assemblies are presented as mean \pm standard deviation. **c,d** Cryo-TEM images of self-assemblies of 0.1 mg/mL following ethanol removal (c) and assemblies annealed at

50 °C for 10 min (d). e,f Cryo-TEM images of self-assemblies of 1 mg/mL following ethanol removal (e) and assemblies annealed at 50 °C for 10 min (f). Scale bars are 200 nm.

“To assess the potential concentration effects, the self-assembly concentration was systematically varied from 0.5 to 1.0 and 0.1 mg/mL. These changes had minimal effect on both the morphology of self-assemblies and the MVV-to-cubosome transition upon annealing at 50 °C (Supplementary Fig. 7).” was added to the revised **Manuscript**.

Fig. 5 to Reviewer #1 and related description were added to **Supplementary Information**.

4. Thermodynamic Stability Measurements

The morphological transitions from MVV (multivesicular vesicles) to sponges and then to cubosomes are discussed only in a conceptual manner. Can calorimetric (e.g., DSC) or spectroscopic (e.g., IR) methods be used to assess the relative stability of each morphological state? Such data would help confirm whether these phase transitions stem from meaningful thermodynamic differences versus purely kinetic traps.

Reviewer 1, Answer point 4: We thank the reviewer for this insightful comment.

As suggested, we tried to use differential scanning calorimetry (DSC) to see if any heat change could be detected during the morphological transitions of self-assemblies (**Fig. 6 to Reviewer #1**). The self-assembly sample in water was measured with blank MQ water used as reference. Unfortunately, no noticeable thermal peaks could be observed in the heating and cooling cycle.

Fig. 6 to Reviewer #1 | Differential scanning calorimetry (DSC) curves of self-assemblies in the first heating and cooling cycle.

As discussed in **Point 2**, for structures at R.T.: cubosomes are the thermodynamic state, while MVVs and sponges are kinetically trapped states, residing in local energy minima. To further investigate the stability of each morphological state, each state was prepared and their stability at R.T. was tracked by LS and cryo-TEM (**Fig. 7 to Reviewer #1**). Specifically, MVVs were

obtained from the original self-assembly (Fig. 2c), sponges were obtained from annealing at 37 °C for 10 min (Fig. 2d), and cubosomes were acquired from annealing at 50 °C for 10 min (Fig. 3d). Cryo-TEM and LS measurements did not reveal any morphological changes of each state after aging for up to seven days, indicating that each resides in a local energy minimum.

Fig. 7 to Reviewer #1 | Stability of the kinetically trapped MVVs and Sponges, and the thermodynamic cubosomes at R.T. **a** D_h , PDI, and count rates measured by LS of self-assemblies solutions over time for the kinetically trapped and thermodynamic products. **b–d** Representative cryo-TEM images of the kinetically trapped MVVs (**a**) and sponges (**b**), and the thermodynamic cubosomes (**c**) aged for 7 d at R.T. Scale bars are 200 nm.

“All four states remained stable over several days at R.T., indicating that each resides in a local energy minimum (Supplementary Fig. 13e and Supplementary Fig. 14).” was added to the revised Manuscript.

Fig. 7 to Reviewer #1 and related description were added to Supplementary Information.

5. Rationale for the 37 °C Experiments

The manuscript describes several experiments at 37 °C, possibly reflecting an interest in physiological conditions. If that is indeed the motivation, it should be explicitly stated, as it would contextualize the choice of temperature.

Reviewer 1, Answer point 5: Thank the reviewer for this important comment. Indeed, the investigation at body temperature of 37 °C brought insights and attention to the energy states of self-assemblies of Janus dendrimers, which have been shown great potential in biomedical applications such as antibacterial nanoreactor⁵ or targeted mRNA delivery⁶⁻⁸. On the other hand, the rationale for two main annealing temperatures of 37 °C and 50 °C in the study was selected according to the noticeable transition points in the temperature sweep by LS (**Fig. 8 to Reviewer #1**).

Fig. 8 to Reviewer #1 | LS characterization of self-assemblies in the absence of ethanol. Relative changes of (a) D_h (%), (b) PDI (%), and (c) derived count rates (%) of assemblies during two heating/cooling cycles in the LS temperature trend measurements.

Fig. 8 to Reviewer #1 shows the relative change of D_h , PDI, and derived count rates of assemblies to the starting points (indicated by the symbol of pentagram) during the heating and cooling cycles. In the first heating cycle, we can identify 37 °C, which was characterized by a slight increase in size and a significant decrease in PDI. 50 °C was also noticed in the first heating cycle, which was characterized by a slight decrease in size and an increase in count rates. These changes suggest a potential transition in the states of the assemblies, which was further investigated by LS, NTA, and cryo-TEM in the kinetic studies (**Fig. 2** and **Fig. 3**).

Fig. 8 to Reviewer #1 and related discussion were added to the revised **Manuscript** and **Supplementary Information**.

6. Bio-Application Claims

While the introduction and conclusion mention potential biomedical applications of these dendrimer assemblies, the system's high sensitivity to ethanol (and possibly other additives) suggests it may not yet be straightforward to translate into complex biological environments. If bio-applications are a genuine aim, demonstration of assembly formation in relevant buffer

solutions would be much more convincing. If not, I don't think the authors need to describe anything related to bio-applications.

Reviewer 1, Answer point 6: Apart from the insights obtained from the current study from a fundamental perspective, the Janus dendrimer self-assemblies constructed from the same/similar molecular structures were demonstrate to possess great potential in various biomedical applications, such as mRNA delivery systems⁶⁻⁸, antimicrobial nanoreactors⁵, and bioimaging⁹.

1) We believe it would be beneficial to bring the messages from our study to the community that careful characterizations of the assembly system are important. The dynamic dendrimer assemblies could reside in energy states with distinct morphologies in response to the thermal history and biological environment, which may pose effects in the applications.

2) On the other hand, the different energy states of the assemblies expand the function possibilities as compared with the static thermodynamically stable system¹⁰. This can advance the applications of Janus dendrimers.

Therefore, we would like to keep the biomedical application context because of the relevance and interest to researchers in the field.

Reviewer #2:

The study explores Janus dendrimer self-assemblies, which can transition reversibly between lamellar vesicles and inverse cubic structures, featuring a complex energy landscape driven by temperature-triggered non-covalent interactions. These assemblies are characterized by a variety of structures, including vesicles, tubular vesicles, sponges, and cubosomes. The energy landscapes are modulated by molecular packing, interdigitation of OEG chains, and hydrogen bonding, with temperature acting as a key factor in shifting between different energy states and affecting packing parameters and assembly pathways. The transition from lamellar to inverse cubic structures is achieved by controlling the packing parameter. The Janus dendrimers provide a versatile system with applications in biomedicine, catalysis, and nanoreactor design. The study highlights the dynamic nature of self-assemblies, which can be kinetically trapped into metastable states. A deeper understanding of these molecular interactions may aid in advancing self-assembly techniques for diverse applications, and the findings propose that these dynamic systems could mimic complex cellular behaviors, offering new insights into supramolecular chemistry. The cryo TEM images are impressive, however, certain aspects of the manuscript needs clarification and supporting data. Thus, I recommend major revision of the manuscript after addressing the following points.

We thank the reviewer for the encouraging comments on our work. We have incorporated the helpful suggestions into our revised manuscript and endeavored to extensively address all the insightful comments below.

Comments:

1. What is the significance of the annealing temperature of 37 °C and 50 °C in this context, considering that oligo(ethylene glycol) (OEG) units exhibit lower critical solution temperature (LCST) behavior? What is the LCST of the present molecules? Is the annealing temperature above or below the LCST of OEG, and how does this influence the self-assembly process?

Reviewer 2, Answer point 1: Thank the reviewer for the comment.

The two main annealing temperatures of 37 °C and 50 °C in the study was selected according to the noticeable transition points in the temperature sweep by LS (**Fig. 1 to Reviewer #2**).

Fig. 1 to Reviewer #2 | LS characterization of self-assemblies in the absence of ethanol. Relative changes of (a) D_h (%), (b) PDI (%), and (c) derived count rates (%) of assemblies during two heating/cooling cycles in the LS temperature trend measurements.

Fig. 1 to Reviewer #2 shows the relative change of D_h , PDI, and derived count rates of assemblies to the starting points (indicated by the symbol of pentagram) during the heating and cooling cycles. In the first heating cycle, we can identify 37 °C, which was characterized by a slight increase in size and a significant decrease in PDI. 50 °C was also noticed in the first heating cycle, which was characterized by a slight decrease in size and an increase in count rates. These changes suggest a potential transition in the states of the assemblies, which was further investigated by LS, NTA, and cryo-TEM in the kinetic studies (**Fig. 2** and **Fig. 3**).

Fig. 1 to Reviewer #2 and related discussion were added to the revised **Manuscript** and **Supplementary Information**.

“LCST investigation”: The molecules that we used were composed of triethylene glycol monomethyl ether (EG₃-OCH₃). According to literatures^{11,12}, the LCST of EG₃-OCH₃ is ~70 °C which is beyond the current studied temperatures.

We further used UV-Vis to monitor the turbidity of the self-assembly sample in the temperature range of interest (from 20 °C to 50 °C). If the molecules exhibited LCST behavior, a drastic decrease should be expected in the transmittance of the sample.

Fig. 2 to Reviewer #2 | Optical transmittances of (a) Janus dendrimer self-assemblies in the absence of ethanol and (b) EG₃-OCH₃ solution (0.1 wt%) as a function of temperature. The transmittances were recorded at a wavelength of 600 nm with a UV–vis spectrometer.

The transmittance of the self-assembly sample showed a gradual decrease starting around 35 °C (**Fig. 2a to Reviewer #2**). We could not attribute this gradual decrease to the presence of LCST because morphological transition from MVVs to inverse sponges and cubosomes took place as the temperature increased. The difference in turbidity could be attributed to the different states of self-assemblies. We next decided to check the transmittance of EG₃-OCH₃ solution as a function of time. The transmittance of EG₃-OCH₃ solution was found to be constant, confirming that the LCST of EG₃-OCH₃ was not falling in the investigated temperature range (**Fig. 2b to Reviewer #2**).

From all these results, the annealing temperatures from the current study were below the LCST of EG₃-OCH₃ and the LCST of of EG₃-OCH₃ is not responsible for the observed morphological transitions.

2. LCST is crucial to say further about the kinetic states. The authors may consider recording temp dependent UV-vis spectra to follow turbidity. Additionally, the studies were conducted at temperatures of 37 °C and 50 °C. What is about performing particle size analysis using Light Scattering (LS), Nanoparticle Tracking Analysis (NTA), and Cryo-TEM at RT as it was anyway annealed to RT.

Reviewer 2, Answer point 2: Thank the reviewer for this comment, allowing us to improve the Figures for clarification.

There seems to be some misunderstanding regarding the annealing procedure and the following characterization. Analysis carried out with LS, NTA, and cryo-TEM in **Figs. 2–4** was actually performed with samples at R.T., after the annealing procedure. The size, PDI, particle concentration, and morphologies in **Figs. 2–4** were characterized at R.T. after annealing samples at 37 °C or 50 °C for certain periods of time to investigate the kinetics during the annealing procedure.

We understand that the misunderstanding might be caused by the missing information on the annealing process and characterization. To further clarify this, we added two schematic figures (**Fig. 3 to Reviewer #2** and **Fig. 4 to Reviewer #2**), illustrating the annealing process at 37 °C and 50 °C and the following characterization of self-assemblies. And we hope these two added figures could provide readers with the necessary information.

Fig. 3 to Reviewer #2 | Annealing process at 37 °C and characterization of self-assemblies.

Fig. 4 to Reviewer #2 | Annealing process at 50 °C and characterization of self-assemblies.

Fig. 3 to Reviewer #2 and **Fig. 4 to Reviewer #2** were added as **Fig. 2a** and **Fig. 3a** in the revised **Manuscript**.

“To further investigate the annealing effects on the self-assemblies at 37 °C and 50 °C (heating to 37 °C or 50 °C for certain periods of time, then cooling down to room temperature (R.T.)), a range of analytical techniques was employed (Fig. 2a).” was added to the revised **Manuscript**.

3. Also, such kinetic state formation also depends on the rate of annealing. However, I could not find such different annealing kinetics to compare the states. It is crucial to investigate the impact of the cooling rate on the morphological transition, as only two cooling conditions (6 °C/min and rapid quenching using ice bath or liquid nitrogen) have been considered. Therefore, it is recommended to cool the system at a slower rate, such as 1 °C/min, to assess the effect of a more gradual cooling process on the transition dynamics.

Reviewer 2, Answer point 3: We thank the reviewer for the helpful comments. As suggested, we performed an additional cooling experiment from 50 °C at a slower rate of 1 °C/min.

Fig. 5 to Reviewer #2 | Effect of cooling speed at 1 °C/min from 50 °C to R.T. on the energy states of self-assemblies observed at R.T. **a** Representative cryo-TEM image of self-assemblies annealed from 50 °C at a speed of 1 °C/min. Scale bar is 200 nm. **b** Quantitative analysis of assembly populations annealed from 50 °C at a speed of 1 °C/min. Images from multiple areas were analyzed to minimize the error ($n > 600$ particles).

As shown in **Fig. 5 to Reviewer #2**, the self-assemblies showed a similar transition from MVVs to cubosomes (approximately 80% in population) as those observed in experiments with 6 °C/min cooling (**Fig. 6e** and **Supplementary Fig. 13b**) and relatively rapid quenching methods using room-temperature air (**Fig. 3d**) or 0 °C ice-water (**Supplementary Fig. 13c**). This demonstrates that an extended cooling process would not pose significant effect on the morphological transitions.

The result of the experiment with an additional cooling rate at 1 °C/min was included in the revised **Manuscript**:

“The transition to cubosomes occurred consistently upon cooling from 50 °C, whether at a slow rate of 1 °C/min (**Supplementary Fig. 13a**), a moderate rate of 6 °C/min (**Fig. 6e** and **Supplementary Fig. 13b**), or via rapid air quenching to R.T. (**Fig. 3e**).”

Supplementary Fig. 13 was modified accordingly by adding the result from cooling rate of 1 °C/min.

4. Based on the temperature, the polymeric nanostructures varies. However, how that correlates to the packing parameter was not clear. No discussion was attempted as well.

Reviewer 2, Answer point 4: Thank the reviewer for the helpful comment that allowed us to improve the manuscript.

The distinct morphologies of self-assemblies, from lamellar vesicles to inverse sponges and cubosomes, helped to set a clear indication of the packing parameter (p) of individual structure. Specifically: MVVs (lamellar phase, $0.5 < p < 1$), intermediate sponges (inverse phase, $p > 1$), cubosomes (inverse phase, $p > 1$), MLVs (lamellar phase, $0.5 < p < 1$).

We have included this information in the discussion on the relationship between molecular packing parameters and the energy states of structures in the section of “**Energy landscapes navigated by molecular design**” of the revised **Manuscript**.

5. Why the control molecule without the methoxy was taken. The additional hydrogen bonding might change the kinetic states, energetics and lead to different LCST behaviors. The relevant discussion was also condensed in a small paragraph. Detailed experiments with the control molecules just like the main molecules might show the differences.

Reviewer 2, Answer point 5: The purpose for the demonstration with the newly synthesized molecules was to show that self-assemblies could be guided to new energy states by molecular design. This was realized by the introduction of the additional hydrogen bonding to detain the lamellar structures in the investigated temperature range. As nicely summarized by a comment from Reviewer 3, we were able to strongly demonstrate that the redesigned molecular interactions using molecules with –OH end groups could reshape the energy landscape.

Briefly, the molecular mechanism for the formation of the kinetically trapped MVVs (lamellar phase, $0.5 < p < 1$) during the injection process was the partial interdigitation of the OEG chains (**Supplementary Fig. 15a–c**). The OEG interdigitation helped to stabilize the MVVs in a local energy minimum as the kinetic state. Upon increasing the temperature during annealing, this weak interdigitation among OEG chains was disrupted, allowing the molecules revert to their favorable configuration with $p > 1$, triggering the transition from MVVs to inverse cubosomes. Could the interdigitation be stabilized to retain the lamellar MVVs during the annealing? Then the self-assemblies could achieve a new energy landscape as compared with the main molecules. To achieve this, the –OCH₃ end groups of the dendrimer molecules were replaced with –OH. Our previous study showed that the introduced hydrogen bonding in the interdigitation could stabilize the interdigitation of OEG up to 60 °C¹³. The same annealing process was then performed with the newly assembled samples. By carefully checking the morphologies of assemblies at various temperatures, the assemblies displayed lamellar vesicular structures due to the locked H-bonding interdigitation within the OEG units (**Fig. 7**). Therefore, we successfully showed that a new energy landscape consisted of lamellar vesicles could be achieved via molecular design.

Here, we do not expect the new molecules with –OH end groups would show any LCST behaviors in the studied temperature range (R.T. to 50 °C). Similar as other thermos-responsive molecules, the LCST of oligo-ethylene glycol increases with the hydrophilicity of the molecules¹¹. The LCST of the more hydrophilic new molecules (if any) will be higher than the main molecules with –OCH₃ end groups, which has not been detected in the studied temperature range (**Fig. 2 to Reviewer #2**).

Detailed relevant discussions, including the rationale for the molecular design with –OH end groups, was presented in the first two paragraphs in the section of “**Energy landscapes navigated by molecular design**” in the **Manuscript** (Lines 358-397).

6. SAXS data in the ESI was qualitative. Can it be used to distinguish between the polymeric structures or the ordering?

Reviewer 2, Answer point 6: Thank the reviewer for the valuable comment.

The small-angle X-ray scattering (SAXS) profiles can be used to characterize the self-assembly structures and the ordering. A clear morphological transformation from a lamellar to a non-lamellar phase is evident from the distinct SAXS patterns (**Supplementary Fig. 5**). By fitting the SAXS profile of the multilamellar vesicles (MVVs) to an appropriate structural model, quantitative parameters were extracted, as detailed in the revised **Supplementary Information (Supplementary Table 1)**. In contrast, the SAXS profile of the inverse cubosomes lacks higher-order Bragg reflections, indicating limited long-range order and precluding a similarly detailed structural analysis. We attribute this lack of long-range ordering to the dynamic nature of the cubosomes, a feature reported in other complex systems¹⁴.

Quantitative structural analysis on MVVs, detailed fitting model and related description were added to the revised **Supplementary Information**.

7. In line 126, figure number of supplementary information is wrong.

Reviewer 2, Answer point 7: We would like to thank the reviewer for pointing out the mistake. The figure numbers in **Supplementary Fig. 3** have been corrected.

8. ESI: Page 3, Scheme 1, molecular structures of 8 is to be corrected.

Reviewer 2, Answer point 8: We checked the molecular structure of **8** in **Supplementary Scheme 1**. And it agrees with molecules with same/similar components in literatures^{3,15} and is correct in our opinion. We would be grateful if the reviewer could give further information on the mistake. We apologize for the inconvenience in advance.

Reviewer #3:

The manuscript by Luan and co-workers presents an investigation into kinetically controlled supramolecular systems, utilizing temperature-driven transitions to navigate complex energy landscapes. By employing lamellar vesicles and inverse cubic structures, the authors demonstrate how amphiphilic moieties capable of hydrogen bonding enable dynamic pathway selection. A particularly striking aspect of this study is the clear demonstration of thermal history dependence, where the structural transitions are dictated by the applied temperature and solvent conditions. The authors employ a different set of microscopic and spectroscopic techniques to characterize their systems, offering important insights into how molecular packing, flexibility, and non-equilibrium self-assembly govern structural evolution. Their detailed discussion on the interplay between packing parameters and reconfigurable energy landscapes provides a compelling framework for understanding the design rules of supramolecular architectures. This work effectively bridges supramolecular chemistry, non-equilibrium control, and adaptive self-assembly, with promising implications for biological applications. Moreover, it opens alternative directions in the study of the origins of life, where transient assemblies could have emerged and persisted under temperature gradients, forming dynamic structures with defined lifetimes. Overall, the manuscript represents an important contribution to the field. I recommend publication in Nature Communications, pending the requested revisions to clarify specific mechanistic aspects and improve readability in certain sections.

We thank the reviewer for the encouraging comments on our work and endorsement of its suitability for inclusion in Nature Communications. We have incorporated the helpful suggestions into our revised manuscript and endeavored to extensively address all the insightful comments below.

1) The study centers around temperature-driven assembly pathways, with detailed comparisons between 37 °C and 50 °C. Could the authors clarify the rationale behind the selection of these specific temperatures? Additionally, it would be informative to understand whether the system displays any notable behavior in the intermediate temperature range, as this could provide further insight into the transition kinetics and possible intermediate states.

Reviewer 3, Answer point 1: Thank the reviewer for bringing up this important comment and the additional suggestion.

The rationale for the two main annealing temperatures of 37 °C and 50 °C in the study was selected according to the noticeable transition points in the temperature sweep by LS (**Fig. 1 to Reviewer #3**).

Fig. 1 to Reviewer #3 | LS characterization of self-assemblies in the absence of ethanol. Relative changes of (a) D_h (%), (b) PDI (%), and (c) derived count rates (%) of assemblies during two heating/cooling cycles in the LS temperature trend measurements.

Fig. 1 to Reviewer #3 shows the relative change of D_h , PDI, and derived count rates of assemblies to the starting points (indicated by the symbol of pentagram) during the heating and cooling cycles. In the first heating cycle, we can identify 37 °C, which was characterized by a slight increase in size and a significant decrease in PDI. 50 °C was also noticed in the first heating cycle, which was characterized by a slight decrease in size and an increase in count rates. These changes suggest a potential transition in the states of the assemblies, which was further investigated by LS, NTA, and cryo-TEM in the kinetic studies (**Fig. 2** and **Fig. 3**).

Fig. 1 to Reviewer #3 and related discussion were added to the revised **Manuscript** and **Supplementary Information**.

Intermediate temperature annealing: To investigate and compare the transition kinetics in the intermediate temperature range between 37 °C and 50 °C, we performed a kinetic investigation of self-assemblies via annealing at 44 °C (heating to 44 °C for certain periods of time, then cooling down to R.T.) in the absence of ethanol (**Fig. 2 to Reviewer #3**).

Fig. 2 to Reviewer #3 | Kinetic investigation of self-assembled Janus dendrimers via 44 °C annealing in the absence of ethanol. **a** Annealing process at 44 °C and characterization of self-assemblies. **b** Relative change of D_h (%), PDI (%), and derived count rates (%) measured by LS as a function of annealing time at 44 °C. **c–f** Cryo-TEM images of self-assemblies annealed at 44 °C for 10 min (**c**), 4 h (**d**), 9 h (**e**), and 28 h (**f**). Scale bars are 200 nm. **g** Quantitative analysis of assembly populations after ethanol removal and post-annealing at 44 °C (**c–f**). For each condition, images from multiple areas and batches were analyzed to minimize the error ($n > 1100$ particles per condition).

After annealing treatment at 44 °C, the self-assemblies showed an intermediate transition kinetics between 37 °C and 50 °C, which corresponds well with the temperature series (**Fig. 2 to Reviewer #3**). Specifically, 44 °C annealing showed similar trends but slower transition kinetics in size (D_h), PDI, and count rates in LS with results obtained from 50 °C annealing (**Fig. 2b to Reviewer #3** and **Fig. 3b**). The morphological transitions were further confirmed by cryo-TEM during the annealing process at 44 °C. Similar to transitions observed at 37 °C and 50 °C, vesicles gradually transformed to sponges and finally cubosomes (**Fig. 2c–f to**

Reviewer #3). The transition completed after 28 h of annealing, reaching approximately 80% of cubosomes (**Fig. 2g to Reviewer #3**).

We have identified two pathways for the transition from original vesicles (MVs) to cubosomes (see updated **Fig. 6g** in Manuscript). Although the transition kinetics of 44 °C annealing (completion time: 28 h) falls in the middle of 37 °C (completion time: 72 h) and 50 °C (completion time: 10 min) annealing, it remains unknown which pathway 44 °C annealing followed. We will now try to address this in the following point.

2) The manuscript highlights an interesting interplay between temperature and annealing time in directing distinct assembly pathways. While the difference in structural outcomes at 37 °C (72 h) and 50 °C (10 min) is striking, it remains unclear whether time alone at intermediate temperatures could yield similar transitions. Could the authors elaborate on whether prolonged annealing at lower temperatures might eventually access the same cubosome phase as rapid heating to 50 °C? Conversely, is there a critical temperature–time threshold beyond which pathway selection becomes irreversible or biased?

Reviewer 3, Answer point 2: Thank the reviewer for this comment. There seems to be some misunderstanding regarding to the structural outcomes for 37 °C and 50 °C annealing. We will try to elaborate on this.

As summarized in **Fig. 5a,b**, the transitions from vesicles to cubosomes exhibited a much faster kinetics when annealing was performed at 50 °C (completion after 10 min) than at 37 °C (completion after 72 h). Apart from the difference in transition kinetics, annealing at both temperatures resulted in the same final structures of cubosomes (R.T. III, **Fig. 6d**). The slow transition kinetics of 37 °C annealing allowed the acquisition of kinetically trapped state of sponges, which is an intermediate form en route to cubosomes (R.T. II, **Fig. 6c**). Therefore, prolonged annealing resulted in the cubosome structures, suggesting cubosomes are the thermodynamically stable state at R.T. It is noteworthy that although the final structure is the same, the MVs underwent two distinct pathways to transform into cubosomes, as highlighted in **Fig. 6g**:

Pathway 1, 37 °C annealing: MVs (lamellar phase, at R.T.) → sponges/cubosomes (inverse phase, at 37 °C and R.T.);

Pathway 2, 50 °C annealing: MVs (lamellar phase, at R.T.) → sponges/cubosomes (inverse phase, at 37 °C) → MLVs (lamellar phase, at 50 °C) → cubosomes (inverse phase, at R.T.).

A question immediately raised: which pathway would the intermediate annealing temperature (for example 44 °C annealing) follows? Furthermore, what is the pathway selection temperature between Pathway 1 and 2?

To provide information to these questions, we investigated the morphological state at intermediate temperature of 44 °C and at a higher temperature than 50 °C (here, we chose 60 °C, which is the highest temperature that could be obtained with the Vitrobot vitrification system for cryo-TEM sample preparation).

Fig. 3 to Reviewer #3 | Pathway selections of self-assemblies via 44 °C and 60 °C annealing. **a–d** Cryo-TEM images of self-assemblies vitrified at the indicated temperatures during the heating/cooling cycle. Self-assemblies were vitrified after being equilibrated at 44 °C and 60 °C for 35 min and 10 min, respectively. Scale bars are 200 nm.

Annealing self-assemblies at 44 °C followed the same pathway with 37 °C annealing (Pathway 1, **Fig. 6g**) as inverse sponges/cubosomes were found at 44 °C (**Fig. 3a,b to Reviewer #3**). When the self-assemblies were annealed at 60 °C, the transition followed Pathway 2 with the formation of the lamellar MLVs structures at 60 °C (**Fig. 3c to Reviewer #3**). Combining all these results, we can identify a critical pathway selection temperature of 50 °C of the transitions from MVVs to cubosomes.

We have further clarified this point by updating the **Fig. 6g** and included relevant discussion in the revised **Manuscript**.

3) Given the elegant demonstration of structural transitions driven by temperature and thermal history, have the authors considered using confocal or super-resolution microscopy techniques to capture these dynamics in real time? While the final structures may be below the resolution limit for direct observation, such approaches could potentially offer valuable insight into fusion events, morphological transitions, or heterogeneity within the population—especially during early stages of annealing. Even partial visualization might complement the cryo-TEM and scattering data, offering a richer picture of the system’s kinetic behavior.

Reviewer 3, Answer point 3: It would be interesting to capture the morphological transition using microscopy. However, in our opinion, it is challenging to capture and distinguish the distinct assembly structures involved in this study using microscopy at the current stage, because:

a) The characteristic size of self-assemblies is around 200 nm. This is far below the resolution of confocal microscopy.

b) For super-resolution microscopy, imaging samples in solution is challenging and it requires the fixation of the particles. This inevitably affects the diffusion ability of molecules and could affect the fusion between particles as observed during the morphological transition.

c) Assuming that the particles could be imaged, it is challenging to distinguish the internal structure difference, as the nanochannels (channel diameters below 10 nm) structure of cubosomes were too small to be detected.

In light of the suggestion, we tried to use super-resolution microscopy to visualize MVVs (original self-assembly sample) and cubosomes (annealed from 50 °C) loaded with fluorescent molecular probes (pyrene-Alexa Fluor 647, Py-AF₆₄₇). Py-AF₆₄₇ probes were successfully loaded on the surface of the self-assemblies using an insertion method reported by our group^{16,17}. The stochastic activation of individual fluorophores enabled us to visualize the distribution of Py-AF₆₄₇ on the self-assemblies at the single-probe level^{17,18}. As expected, the structural difference of cubosomes and MVVs cannot be distinguished due to the limitation of the current technology (**Fig. 4 to Reviewer #3**).

Fig. 4 to Reviewer #3 | Super-resolution fluorescent images of MVVs (a) and cubosome (b) loaded with the fluorescent molecular probes (Py-AF₆₄₇). The excitation wavelength is 640 nm. Scale bars are 200 nm.

Nevertheless, cryo-TEM is a very powerful technique to preserve and image the native state of samples at specific temperatures. And we imaged a sufficiently large amount of structures ($n > 500$ for each condition) from randomly chosen areas or even different batches, which helped us to obtain quantitative analysis of the percentage of structures with different morphologies. Together with other characterization techniques such as DLS and NTA, the kinetics of the system were well presented.

4) The manuscript compellingly shows how both annealing temperature and solvent influence transition kinetics and particle dimensions, particularly highlighting ethanol's role in enhancing cubosome growth via apparent fusion processes. While one might be tempted to ask about changing the solvent system altogether, this is not the intent here. Rather, could the authors elaborate on how ethanol subtly reshapes the energy landscape—possibly by modulating hydration, packing parameters, or membrane fluidity? Is there a concentration threshold beyond which fusion becomes strongly favored? A more detailed discussion of how solvent environment and thermal history co-regulate the transition pathways would enrich the mechanistic understanding of this finely tuned supramolecular system.

Reviewer 3, Answer point 4: Thank the reviewer for the comment. Our Janus dendrimer molecules bear similar hydrophobic alkyl chains (C12) and have comparable molecular weight and dynamics as lipids³. The findings on the effect of ethanol on lipid bilayers could be applicable to our molecular system.

a) Ethanol partitions into the hydrophobic interfaces and increases the area per molecule, which leads to an increase in the size of the vesicles^{19,20}. This phenomenon was also observed in our self-assemblies that the size of the self-assemblies was slighter greater with ethanol (**Fig. 4a,b**).

b) The presence of ethanol in principle did not significantly alter the packing parameters of molecules to the extent of forming different types of self-assemblies. Similar to the transitions observed in the absence of ethanol (**Figs. 2,3**), the self-assemblies displayed the same morphological transition pathway: from MVVs (lamellar phase, $0.5 < p < 1$) to inverse sponges (inverse phase, $p > 1$) and finally to cubosomes (inverse phase, $p > 1$) (**Fig. 4**).

c) The fluidity of the lipid membrane was enhanced by ethanol^{21,22}, facilitating the fusion of lipid vesicles²⁰. We observed a similar effect as the size of resulting cubosomes was significantly larger compared to those formed without ethanol after 37 °C annealing (**Fig. 5a,c**).

d) As fusion also occurred in samples without ethanol, we do not expect an ethanol concentration threshold triggering the fusion process. Nevertheless, the presence of ethanol evidently facilitated the fusion due to the enhanced membrane fluidity.

The **Manuscript** has been revised accordingly to include the effect of ethanol on the energy landscape:

“These results indicate that fusion occurred during the transformation during 37 °C annealing, with ethanol significantly enhancing the fusion process due to the enhanced membrane fluidity as shown in lipid systems³⁸.”

“This increase was attributed to the swelling effect of ethanol on the area per molecule, which was also observed in lipid molecules with similar alkyl chains^{19,37,38}.”

5) The manuscript presents a nice demonstration of how different thermal pathways give rise to multiple distinct self-assembled states, including MVVs, sponges, MLVs, and cubosomes. Given the system’s sensitivity to thermal history and cooling rates, it would be interesting to understand its robustness across repeated thermal cycles. Can the authors comment on how many heating/cooling cycles the system can undergo before irreversible changes—such as degradation, kinetic trapping, or loss of responsiveness—begin to occur?

Reviewer 3, Answer point 5: We appreciate the reviewer for the helpful comment. As shown in the updated **Fig. 6g**, we would like to highlight that distinct structures were achieved through the pathway selections of various energy states of the self-assemblies. Specifically, the energy landscape of assemblies at R.T. composed one thermodynamically stable state of cubosomes (R.T. III, **Fig. 6g**) and three kinetically trapped states of MVVs (R.T. I, **Fig. 6g**), sponges (R.T. II, **Fig. 6g**), and MLVs (R.T. IV, **Fig. 6g**).

Fig. 5 to Reviewer #3 | Schematic representation of the energy landscape of self-assemblies of MVV and Cubosome.

The transition from MVVs to cubosomes proceeded energetically downhill to a global minimum at R.T. (**Fig. 5 to Reviewer #3**). This transition is hence typically irreversible. For example, cubosomes obtained from the first annealing treatment remained unchanged in the second annealing cycle, regardless of the annealing at 37 °C or 50 °C (**Supplementary Fig. 6**).

On the other hand, we could reversibly convert the thermodynamically stable cubosomes (inverse phase) back to MLVs (lamellar phase) through a kinetically trapped strategy. Specifically, the packing parameter of the molecules favored a value for MLV formation (lamellar phase, $0.5 < p < 1$) above 50 °C. Through an ultrafast quenching method, the MLVs (lamellar phase) could be kinetically trapped at R.T., allowing a transition from cubosomes back to vesicular structures (R.T. IV, Fig. 6g and Supplementary Fig. 13d).

“Notably, ultrafast quenching method enabled a transition from thermodynamic cubosomes back to kinetically trapped vesicular MLV structures.” was added in the revised **Manuscript**.

6) Given the system’s sensitivity to thermal history and the presence of kinetically trapped states, could the authors comment on whether the age of the sample prior to annealing affects the subsequent assembly behavior? For instance, would allowing the sample to rest at room temperature for extended periods (e.g., hours to days) before initiating thermal treatment influence the accessibility or stability of certain morphologies? Exploring whether pre-annealing aging introduces additional kinetic constraints—or perhaps even facilitates alternative pathways—would provide deeper insight into the temporal plasticity of these supramolecular assemblies.

Reviewer 3, Answer point 6: Thank the reviewer for this helpful comment. As suggested by the reviewer, we examined the effect of extended aging at room temperature (R.T.) for 3 days on the transition of self-assemblies (without ethanol) induced by annealing.

Fig. 6 to Reviewer #3 | Effect of extended aging on the morphological transition of self-assemblies via 50 °C annealing in the absence of ethanol. a R.T. aging and the following

annealing process at 50 °C and characterization of self-assemblies. **b** D_h profiles of self-assemblies before and after aging at R.T. as determined by LS. Ethanol was removed by thorough dialysis. D_h and PDI of the self-assemblies are presented as mean \pm standard deviation. **c,d** Cryo-TEM images of self-assemblies after 3 d aging at R.T. (**c**) and annealed at 50 °C for 10 min (**d**). Scale bars are 200 nm.

After aging the self-assemblies at R.T. for 3 d, there were no significant changes in size and PDI as measured by LS (**Fig. 6b to Reviewer #3**). Meanwhile, the self-assemblies remained to be MVVs (**Fig. 6c to Reviewer #3**), which is identical to the original samples (**Fig. 2d and Supplementary Fig. 3a**). After equilibrating at 50 °C for 10 min and cooling down to R.T., the same cubosomes were formed (**Fig. 6d to Reviewer #3**). We can conclude that the pre-annealing aging at R.T. does not pose effects on the morphological transformation of the self-assemblies from MVVs to cubosomes.

“Similarly, aging self-assemblies at R.T. for 3 days did not pose noticeable impact on the MVV structures as well as their transformation to cubosomes following 50 °C annealing (Supplementary Fig. 8).” was added to the revised **Manuscript**.

Fig. 6 to Reviewer #3 and related description were added to **Supplementary Information**.

7) The redesign with –OH end groups is a strong demonstration of how molecular interactions can reshape the energy landscape. It would be helpful if the authors clearly stated which parameters—such as solvent conditions, presence of ethanol, or other additives—were used in these experiments. For the –OH terminated dendrimers, was any co-solvent involved? Moreover, could the authors comment on the generality of these findings? Are the observed transitions and morphologies specific to this Janus dendrimer design, or might these principles extend to related systems?

Reviewer 3, Answer point 7: Thank the reviewer for the valuable comment, which enabled us to enhance the clarity the manuscript.

The study using the self-assemblies of molecules with –OH end groups was performed following the same self-assembly and dialysis procedures. Therefore, the self-assemblies were examined in the absence of ethanol. This design ensures that the only variable is the molecular end group, allowing us to attribute any observed differences in the self-assembly energy landscape solely to molecular structure without solvent effects.

We have clarified the experimental condition of the new molecules in the revised **Manuscript** and added relevant description in the caption of **Fig. 7**.

We appreciate the reviewer’s insightful question regarding the generality of our findings. Our current study, together with our previous work on related Janus dendrimers¹³, supports the broader applicability of the design principles employed. We believe that the strategy of

exploiting non-covalent interactions in conjunction with dynamic molecular packing parameters to navigate rich energy landscapes of distinct morphologies could be extended to other classes of Janus dendrimers. In light of the reviewer's comment, we added our thinking of the current concept for the applicability on other Janus dendrimer systems in the revised **Manuscript**, as follows:

“Supported by our previous work¹⁹, we expect that exploiting non-covalent interactions and dynamic molecular packing parameters to navigate rich energy landscapes of distinct morphologies provides a generalizable framework applicable to other classes of Janus dendrimers, including Janus glycodendrimers⁴³, ionizable amphiphilic Janus dendrimers⁴⁴, and stereochemical Janus⁴⁵.”

References:

1. Wehner, M., Rohr, M.I.S., Stepanenko, V. & Wurthner, F. Control of self-assembly pathways toward conglomerate and racemic supramolecular polymers. *Nat. Commun.* **11**, 5460 (2020).
2. Wehner, M. et al. Supramolecular Polymorphism in One-Dimensional Self-Assembly by Kinetic Pathway Control. *J. Am. Chem. Soc.* **141**, 6092–6107 (2019).
3. Percec, V. et al. Self-assembly of Janus dendrimers into uniform dendrimersomes and other complex architectures. *Science* **328**, 1009–1014 (2010).
4. Buzzacchera, I. et al. Screening libraries of amphiphilic Janus dendrimers based on natural phenolic acids to discover monodisperse unilamellar dendrimersomes. *Biomacromolecules* **20**, 712–727 (2019).
5. Potter, M. et al. Controlled dendrimersome nanoreactor system for localized hypochlorite-induced killing of bacteria. *ACS Nano* **14**, 17333–17353 (2020).
6. Zhang, D. et al. Targeted delivery of mRNA with one-component ionizable amphiphilic Janus dendrimers. *J. Am. Chem. Soc.* **143**, 17975–17982 (2021).
7. Zhang, D. et al. One-Component Multifunctional Sequence-Defined Ionizable Amphiphilic Janus Dendrimer Delivery Systems for mRNA. *J. Am. Chem. Soc.* **143**, 12315–12327 (2021).
8. Zhang, D. et al. The Unexpected Importance of the Primary Structure of the Hydrophobic Part of One-Component Ionizable Amphiphilic Janus Dendrimers in Targeted mRNA Delivery Activity. *J. Am. Chem. Soc.* **144**, 4746–4753 (2022).
9. Filippi, M. et al. Novel stable dendrimersome formulation for safe bioimaging applications. *Nanoscale* **7**, 12943–12954 (2015).
10. Tantakitti, F. et al. Energy landscapes and functions of supramolecular systems. *Nat. Mater.* **15**, 469–476 (2016).
11. Vancoillie, G., Frank, D. & Hoogenboom, R. Thermoresponsive poly(oligo ethylene glycol acrylates). *Prog. Polym. Sci.* **39**, 1074–1095 (2014).
12. Hua, F., Jiang, X., Li, D. & Zhao, B. Well-defined thermosensitive, water-soluble polyacrylates and polystyrenics with short pendant oligo(ethylene glycol) groups synthesized by nitroxide-mediated radical polymerization. *J. Polym. Sci. Part A Polym. Chem.* **44**, 2454–2467 (2006).
13. Luan, J. et al. Complex Energy Landscapes of Self-Assembled Vesicles. *J. Am. Chem. Soc.* **145**, 15496–15506 (2023).
14. Sagalowicz, L., Acquistapace, S., Watzke, H.J. & Michel, M. Study of Liquid Crystal Space Groups Using Controlled Tilting with Cryogenic Transmission Electron Microscopy. *Langmuir* **23**, 12003–12009 (2007).
15. Percec, V. et al. Modular synthesis of amphiphilic Janus glycodendrimers and their self-assembly into glycodendrimersomes and other complex architectures with bioactivity to biomedically relevant lectins. *J. Am. Chem. Soc.* **135**, 9055–9077 (2013).
16. Zhang, S. et al. Adaptive insertion of a hydrophobic anchor into a poly(ethylene glycol) host for programmable surface functionalization. *Nat. Chem.* **15**, 240–247 (2023).
17. Li, W., Zhang, S., Rijpkema, S.J. & Wilson, D.A. Surface-Area Determination of Anisotropic Polymersomes by Amphiphilic Molecular Probe Loading. *Angew. Chem. Int. Ed.* **62**, e202305795 (2023).
18. ONI, Nanoimager. <https://oni.bio/nanoimager/>.

19. Ly, H.V. & Longo, M.L. The influence of short-chain alcohols on interfacial tension, mechanical properties, area/molecule, and permeability of fluid lipid bilayers. *Biophys. J.* **87**, 1013–1033 (2004).
20. Shobhna & Kashyap, H.K. Deciphering Ethanol-Driven Swelling, Rupturing, Aggregation, and Fusion of Lipid Vesicles Using Coarse-Grained Molecular Dynamics Simulations. *Langmuir* **38**, 2445–2459 (2022).
21. Barry, J.A. & Gawrisch, K. Direct NMR Evidence for Ethanol Binding to the Lipid-Water Interface of Phospholipid Bilayers. *Biochemistry* **33**, 8082–8088 (1994).
22. Gurtovenko, A.A. & Anwar, J. Interaction of Ethanol with Biological Membranes: The Formation of Non-bilayer Structures within the Membrane Interior and their Significance. *J. Phys. Chem. B* **113**, 1983–1992 (2009).

Response Letter

We would like to express our sincere gratitude to the reviewers for their careful evaluations on our revised manuscript. In this Response Letter, we have addressed the remaining comments from reviewers.

A point-by-point response to the reviewers' comments is provided below, where reviewers' comments are marked in black while **our response is in blue**. The revised Manuscript with **changes highlighted in red** has been provided.

Reviewers comments:

Reviewer #1:

I have carefully reviewed the revised manuscript. The authors addressed the reviewers' comments exceptionally thoroughly and sincerely, adding numerous experimental results and providing credible responses to all the concerns raised. I wish to express my sincere commendation to the authors for their diligent efforts. The manuscript aligns well with the scope of Nature Communications, and I strongly recommend publishing it in its current form.

We sincerely thank the reviewer for the thoughtful assessment and generous comments. We greatly appreciate the recognition of our efforts in addressing the feedback and are grateful for the recommendation to publish our manuscript.

Reviewer #2:

I have gone through the authors' responses not only to my comments but also the concerns raised by other reviewers as well. In my view the manuscript has now improved significantly upon incorporating the suggested experiments and quantitative analysis. Further, manuscript has also become easy to follow upon incorporating the schemes.

Thus, in my view manuscript may now be accepted in Nature Comm upon incorporating the following minor aspects:

We sincerely thank the reviewer for the careful evaluation of our revised manuscript and for acknowledging the improvements made in response to the reviewers' suggestions. We greatly appreciate the recognition of the additional experiments, quantitative analysis, and schematic illustrations, which have helped improve the clarity and overall quality of the manuscript.

We are grateful for the feedback and will carefully address the remaining minor points to ensure the manuscript for publication in *Nature Communications*.

Comments:

1. Page 3, Scheme 1, molecular structures of 8 has few bonds missing. Please check if they go missing during PDF conversion.

Reviewer 2, Answer point 1: We ensure all bond representations are intact in the final version of **Supplementary Information**.

2. The author may cite the following recent paper that talks about packing parameter mediated transformation of vesicle to micelles. ChemComm, 2023, 59, 13195 – 13198.

Reviewer 2, Answer point 2: The suggested reference has been cited in the **Manuscript**.

Reviewer #3:

The authors have addressed all of my comments and concerns in their revision. I recommend acceptance of the manuscript in its current form.

We sincerely thank the reviewer for the time and thoughtful assessment. We truly appreciate the positive recommendation for publication.